# LEARNING LABEL ENCODINGS FOR DEEP REGRESSION

**Deval Shah & Tor M. Aamodt**
Department of Electrical and Computer Engineering
University of British Columbia, Vancouver, BC, Canada
`{devalshah,aamodt}@ece.ubc.ca`

## ABSTRACT

Deep regression networks are widely used to tackle the problem of predicting a continuous value for a given input. Task-specialized approaches for training regression networks have shown significant improvement over generic approaches, such as direct regression. More recently, a generic approach based on regression by binary classification using binary-encoded labels has shown significant improvement over direct regression. The space of label encodings for regression is large. Lacking heretofore have been automated approaches to find a good label encoding for a given application. This paper introduces *Regularized Label Encoding Learning (RLEL)* for end-to-end training of an entire network and its label encoding. RLEL provides a generic approach for tackling regression. Underlying RLEL is our observation that the search space of label encodings can be constrained and efficiently explored by using a continuous search space of real-valued label encodings combined with a regularization function designed to encourage encodings with certain properties. These properties balance the probability of classification error in individual bits against error correction capability. Label encodings found by RLEL result in lower or comparable errors to manually designed label encodings. Applying RLEL results in $10.9\%$ and $12.4\%$ improvement in Mean Absolute Error (MAE) over direct regression and multiclass classification, respectively. Our evaluation demonstrates that RLEL can be combined with off-the-shelf feature extractors and is suitable across different architectures, datasets, and tasks. Code is available at `https://github.com/ubc-aamodt-group/RLEL_regression`.

## 1 INTRODUCTION

Deep regression is an important problem with applications in several fields, including robotics and autonomous vehicles. Recently, neural radiance fields (NeRF) regression networks have shown promising results in novel view synthesis, 3D reconstruction, and scene representation (Liu et al., 2020; Yu et al., 2021). However, a typical generic approach to direct regression, in which the network is trained by minimizing the mean squared or absolute error between targets and predictions, performs poorly compared to task-specialized approaches (Yang et al., 2018; Ruiz et al., 2018; Niu et al., 2016; Fu et al., 2018). Recently, generic approaches based on regression by binary classification have shown significant improvement over direct regression using custom-designed label encodings (Shah et al., 2022). In this approach, a real-valued label is quantized and converted to an $M$-bit binary code, and these binary-encoded labels are used to train $M$ binary classifiers. In the prediction phase, the output code of classifiers is converted to real-valued prediction using a decoding function. Moreover, binary-encoded labels have been proposed for ordinal regression (Li & Lin, 2006; Niu et al., 2016) and multiclass classification (Allwein et al., 2001; Cissé et al., 2012). The use of binary-encoded labels for regression has multiple advantages. Additionally, predicting a set of values (e.g., classifiers' output) instead of one value (direct regression) introduces ensemble diversity, which improves accuracy (Song et al., 2021). Furthermore, encoded labels introduce redundancy in the label presentation, which improves error correcting capability and accuracy (Dietterich & Bakiri, 1995).

Finding suitable label encoding for a given problem is challenging due to the vast design space. Related work on ordinal regression has primarily leveraged unary codes (Li & Lin, 2006; Niu et al., 2016; Fu et al., 2018). Different approaches for label encoding design, including autoencoder, random search, and simulated annealing, have been proposed to design suitable encoding for multiclass classification (Cissé et al., 2012; Dietterich & Bakiri, 1995; Song et al., 2021). However, these encodings

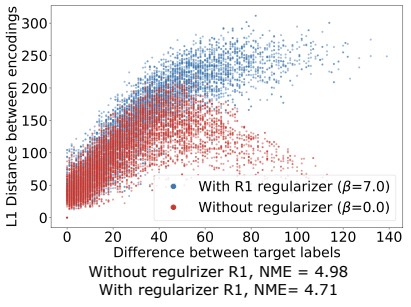 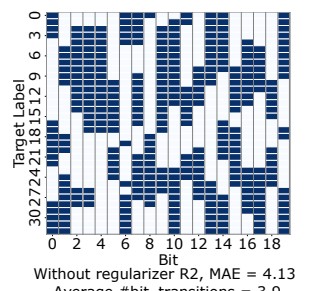 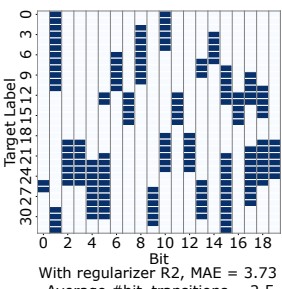

(a) Effect of regularizer R1            (b) Effect of regularizer R2

Figure 1: (a) and (b) demonstrate the effect of proposed regularizers on learned label encodings and the regression error (NME/MAE) for FLD1_s and LFH1 benchmarks (Table 2), respectively. (a) Regularizer R1 encourages the distance between learned encodings to be proportional to the difference between corresponding label values. (b) Regularizer R2 reduces the number of bit transitions per bit, reducing the complexity of decision boundaries to be learned by binary classifiers. Here blue and white colors represent 1 and 0, respectively.

perform relatively poorly for regression due to differences in task objectives (Section 2). More recently, Shah et al. (2022) analyzed and proposed properties of suitable encodings for regression. They empirically demonstrated the effectiveness of manually designed encodings guided by these properties. While establishing the benefits of exploring the space of label encodings for a given task, they did not provide an automated approach to do so.

In this work, we propose *Regularized Label Encoding Learning (RLEL)*, an end-to-end approach to train the network and label encoding together. Binary-encoded labels have discrete search space. This work proposes to relax the assumption of using discrete search space for label encodings. Label encoding design can be approached by regularized search through a continuous space of real-valued label encodings, enabling the use of continuous optimization approaches. Such a formulation enables end-to-end learning of the network parameters and label encoding.

We propose two regularization functions to encourage certain properties in the learned label encoding during training. Specifically, while operating on real-valued label encoding, the regularization functions employed by RLEL are designed to encourage properties previously identified as being helpful for binary-valued label encodings (Shah et al., 2022). The first property encourages the distance between learned encoded labels to be proportional to the difference between corresponding label values, which reduces the regression error. Further, each bit of label encoding can be considered a binary classifier. The second property proposes to reduce the complexity of a binary classifier's decision boundary by reducing the number of bit transitions ($0 \rightarrow 1$ and $1 \rightarrow 0$ transitions in the classifier's target over the range of labels) in the corresponding bit in binarized label encoding.

Figure 1 demonstrates the effect of proposed regularizers on the learned label encodings and regression errors. Figure 1a plots the L1 distance between learned encodings for different target labels versus the difference between corresponding label values. The L1 distance between encodings for distant targets is low without the regularizer. In contrast, the proposed regularizer encourages the learned label encoding to follow the first design property. Figure 1b plots learned label encoding (binarized representation for clarity). Each row represents encoding for a target value, and each column represents a classifier's output over the range of target labels. The use of regularizer R2 reduces the number of bit-transitions (i.e., $1 \rightarrow 0$ and $0 \rightarrow 1$ transitions in a column) to enforce the second design property and consequently reduces the regression error.

We demonstrate that the regularization approach employed by RLEL encourages the desired properties in the label encodings. We evaluate the proposed approach on 11 benchmarks, covering diverse datasets, network architectures, and regression tasks, such as head pose estimation, facial landmark detection, age estimation, and autonomous driving. Label encodings found by RLEL result in lower or comparable errors to manually designed codes and outperform generic encoding design approaches (Gamal et al., 1987; Cissé et al., 2012; Shah et al., 2022). Further, RLEL results in lower error than direct regression and multiclass classification by $10.9\%$ and $12.4\%$, respectively, and even outperforms several task-specialized approaches. We make the following contributions to this work:

**(a) Unary encoding**

| Target Label | Binary-encoded Label | | | | | | |
|---|---|---|---|---|---|---|---|
| | B7 | B6 | B5 | B4 | B3 | B2 | B1 |
| 1 | 0 | 0 | 0 | 0 | 0 | 0 | 0 |
| 2 | 0 | 0 | 0 | 0 | 0 | 0 | 1 |
| 3 | 0 | 0 | 0 | 0 | 0 | 1 | 1 |
| 4 | 0 | 0 | 0 | 0 | 1 | 1 | 1 |
| 5 | 0 | 0 | 0 | 1 | 1 | 1 | 1 |
| 6 | 0 | 0 | 1 | 1 | 1 | 1 | 1 |
| 7 | 0 | 1 | 1 | 1 | 1 | 1 | 1 |
| 8 | 1 | 1 | 1 | 1 | 1 | 1 | 1 |

**(b) Hadamard encoding**

| Target Label | Binary-encoded Label | | | | | | | |
|---|---|---|---|---|---|---|---|---|
| | B8 | B7 | B6 | B5 | B4 | B3 | B2 | B1 |
| 1 | 1 | 1 | 1 | 1 | 1 | 1 | 1 | 1 |
| 2 | 1 | 0 | 1 | 0 | 1 | 0 | 1 | 0 |
| 3 | 1 | 1 | 0 | 0 | 1 | 1 | 0 | 0 |
| 4 | 1 | 0 | 0 | 1 | 1 | 0 | 0 | 1 |
| 5 | 1 | 1 | 1 | 1 | 0 | 0 | 0 | 0 |
| 6 | 1 | 0 | 1 | 0 | 0 | 1 | 0 | 1 |
| 7 | 1 | 1 | 0 | 0 | 0 | 0 | 1 | 1 |
| 8 | 1 | 0 | 0 | 1 | 0 | 1 | 1 | 0 |

**(c) Johnson encoding**

| Target Label | Binary-encoded Label | | | |
|---|---|---|---|---|
| | B4 | B3 | B2 | B1 |
| 1 | 0 | 0 | 0 | 0 |
| 2 | 1 | 0 | 0 | 0 |
| 3 | 1 | 1 | 0 | 0 |
| 4 | 1 | 1 | 1 | 0 |
| 5 | 1 | 1 | 1 | 1 |
| 6 | 0 | 1 | 1 | 1 |
| 7 | 0 | 0 | 1 | 1 |
| 8 | 0 | 0 | 0 | 1 |

**(d) Error distribution**

| Target | B1 |
|---|---|
| 1 | 0 |
| 2 | 0 |
| 3 | 0 |
| 4 | 0 |
| 5 | 1 |
| 6 | 1 |
| 7 | 1 |
| 8 | 1 |

Figure 2: (a-c) Examples of label encodings. Each row represents the binary-encoded target label for a given target. (d) represents the error probability distribution of the classifier-1 for different target values in Johnson encoding. Green lines represent the bit transitions of a classifier.

- We provide an efficient label encoding design approach by combining regularizers with continuous search space of label encodings.

- We analyze properties of suitable encodings in the continuous search space and propose regularization functions for end-to-end learning of network parameters and label encoding.

- We evaluate the proposed approach on 11 benchmarks and show significant improvement over different encoding design methods and generic regression approaches.

## 2 BACKGROUND AND RELATED WORK

This section summarizes relevant background information on regression by binary classification approach and different code design approaches. Task-specific regression approaches are summarized in Appendix A.3. However, a generic regression approach applicable to all tasks is desirable.

### 2.1 REGRESSION BY BINARY CLASSIFICATION

A regression problem can be converted to a set of binary classification subproblems. Prior works proposed to use $N$ binary classifiers for scaled and quantized target labels $\in \{1, 2, ..., N\}$ (Niu et al., 2016; Fu et al., 2018). Here, classifier-$k$'s target output is 1 if the target label is greater than $k$, else 0. Figure 2a represents the target output of binary classifiers for this setup. Shah et al. (2022) proposed *Binary-encoded Labels (BEL)*, a generalized framework for regression by binary classification. In the proposed approach, a real-valued target label is quantized and converted to a binary code $B$ of length $M$ using an encoding function $\mathcal{E}$. $M$ binary classifiers are trained using binary-encoded target labels $B \in \{0, 1\}^M$. During inference, the output of binary classifiers, i.e., predicted code, is converted to the real-valued prediction using a decoding function $\mathcal{D}$.

Using encoded labels introduces error-correction capability, i.e., tolerance to classification error. Hamming distance between two codes (number of differing bits) gives a measure of error-correction capability. Error-correcting codes, such as Hadamard codes (Figure 2b), have been proposed to encode labels in multiclass classification (Dietterich & Bakiri, 1995; Verma & Swami, 2019). BEL showed that such codes are not suitable for regression due to differences in task objectives and classifiers' error probability distribution, and proposed properties of suitable codes for regression.

The first property suggests a trade-off between classification errors and error correction properties. Each classifier learns a decision boundary for *bit transitions* from $1 \rightarrow 0$ and $0 \rightarrow 1$ in the classifier's target bit over the numeric range of labels (green lines within a column in Figure 2). For example, in Johnson encoding (Libaw & Craig, 1953) (Figure 2c), the classifier for bit $B^3$ learns two decision boundaries for bit transitions in intervals $(2, 3)$ and $(6, 7)$. The number of intervals for which the classifier has to learn a separate decision boundary increases with bit transitions, increasing its complexity. Hadamard codes have excellent error-correction properties but have several bit transitions (Figure 2b); this increases the complexity of a classifier's decision boundary and reduces its classification accuracy compared to unary and Johnson codes (Figure 2a and Figure 2c). Rahaman et al. (2019) introduced the term *spectral bias* and demonstrated that neural networks prioritize learning low-frequency functions (i.e., lower local fluctuations). The spectral bias of neural networks provides insights into accuracy improvement with the reduction in the number of bit transitions.

Second, Hamming distance between two codes should increase with the difference between corresponding label values to reduce the probability of making large absolute errors between predicted and target labels. The probability that erroneous predicted code for label X will be decoded as Y decreases as the hamming distance between codes for values X and Y increases. Thus the above rule reduces the regression error. Last, the encoding design should also consider the error probability of classifiers. BEL shows that the error probability of classifiers is not uniform for regression and increases near bit transitions, as shown for classifier $B^1$ in Figure 2d. Here, the probability of predicting 8 for target label 1 is very low, as the bit differing between corresponding codes ($B^1$) has a very low classification error probability. BEL shows that this property can be exploited to design better codes for regression. These three factors significantly affect the suitability of encodings. BEL demonstrates that simple codes sampled based on these properties, such as unary or Johnson code, result in lower errors than widely used error-correcting Hadamard code.

## 2.2 ENCODING DESIGN

Encoding design is a well-studied problem with applications in several fields. Iterative approaches, such as simulated annealing or random walk, have been proposed for code design (Dietterich & Bakiri, 1995; Song et al., 2021). However, iterative approaches are computationally expensive as each iteration requires full/partial training of the network to measure the error for sample encodings. Works on multiclass classification using binary classifiers have demonstrated the effectiveness of error-correcting codes such as Hadamard or random codes (Verma & Swami, 2019; Dietterich & Bakiri, 1995). Cissé et al. (2012) proposed an autoencoder-based approach to design compact codes for multiclass classification problems with a large number of classes. However, these approaches do not consider the task objective and classifiers' nonuniform error probability distribution for regression.

Deep hashing approaches aim to find binary hashing codes for given inputs such that the hashing codes preserve the similarities in the inputs space (Luo et al., 2022; Wang et al., 2018; Xia et al., 2014; Jin et al., 2019; Liu et al., 2016). Deep supervised hashing approaches use the label information to design the loss function. In deep hashing, loss functions are designed to decrease the hamming distance between binary codes for similar images (e.g., same label). In contrast, label encoding design for regression aims to reduce the error between decoded output codes and target labels. Further, deep hashing approaches are designed for classification datasets and do not account for the nonuniform error probability distribution of classifiers observed in regression. As shown in prior work (Shah et al., 2022), classifiers' nonuniform probability significantly affects the design of suitable codes for regression. Thus, a naive adaptation of deep hashing approaches for regression problems performs poorly compared to codes designed by the proposed approach RLEL (Section A.1.5).

## 3 REGULARIZED LABEL ENCODING LEARNING

Regression aims to minimize the error between target labels $y_i$ and predictions $\hat{y}_i$ for a set of training samples $i$. In regression by binary classification, the network learns $M$-bit binary-encoded labels $B_i \in \{0, 1\}^M$. During inference, the predicted code $\hat{B}_i$ is decoded to a real-valued label $\hat{y}_i$. We propose to relax label encodings' search space from a discrete ($\{0, 1\}^M$) to a continuous space ($\mathbb{R}^M$), enabling the use of traditional continuous optimization methods. We propose regularizers to enable efficient search through this space. This work automates the search for label encoding using an end-to-end training approach that learns the network parameters and label encoding together.

This section explains the proposed label encoding learning approach RLEL. First, we explain the regression by binary classification formulation used in this work for end-to-end training of network parameters and label encoding. Further, we introduce properties of suitable label encodings in continuous space. Lastly, we explain the proposed regularizers and loss function that accelerate the search for label encodings by encouraging learned label encoding to exhibit the proposed properties.

### 3.1 LABEL ENCODING LEARNING

**Preliminaries:** Figure 3 represents the formulation used in this work for label encoding learning. $x_i$ and $y_i$ represent the input and the real-valued target label for sample $i$, respectively. We assume $y_i \in [1, N]$ for simplicity as the real-valued targets with any arbitrary numeric range can be scaled and shifted to this range. $Q_i \in \{1, 2, ..., N\}$ represents the quantized target label. The input $x_i$ is

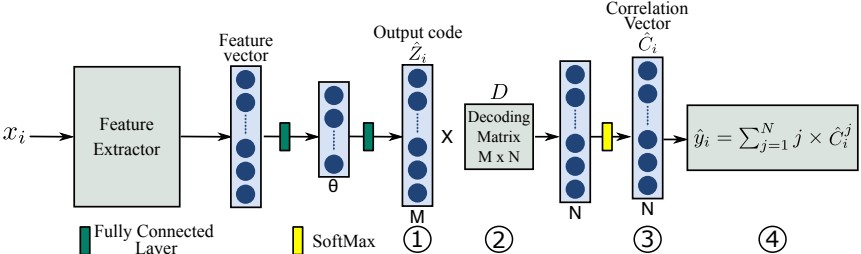

Figure 3: The flow for combined training of feature extractor and label encoding.

Table 1: Summary of notations used in this work

| Notation | Description |
|---|---|
| $x_i, y_i, Q_i$ | Input, real-valued target label, and quantized target label for training example $i$. $y_i \in [1, N]$ and $Q_i \in \{1, 2, ..., N\}$ |
| $N$ | The range of target labels $y_i$; Number of quantization levels for $Q_i$ |
| $M$ | Number of bits/values for label encoding |
| $B_i, \hat{B}_i$ | Target and predicted binary-encoded labels (used for hand-crafted label encoding |
| $\hat{Z}_i$ | Predicted real-valued encodings; activation values of the output code layer in Figure 3 |
| $E$ | Learned label encoding through RLEL; calculated from $\hat{Z}_i$ for all training examples using equation 1 |
| $\hat{C}_i$ | Output correlation vector of length $N$. Here $\hat{C}_i^j$ gives a measure of the probability that predicted label value is equal to $j$ |
| $D$ | Decoding matrix that converts the predicted encodings to a correlation vector $\hat{C}_i$ |

passed through a feature extractor and fully connected (FC) layers to generate the predicted encoding $\hat{Z}_i \in \mathbb{R}^M$ ①. Here, an FC layer of size $\theta$ ($\theta < M$) is added between the feature vector and output code. This layer reduces the number of parameters in FC layers and improves accuracy, as shown by previous work (Shah et al., 2022). Each neuron of the output code is a binary classifier, and the magnitude $\hat{Z}_i^k$ gives a measure of the confidence of the classifier-$k$ (Allwein et al., 2001). The output code and a decoding matrix $D \in \mathbb{R}^{M \times N}$ are multiplied ②, and the output is passed through a softmax function to give a correlation vector $\hat{C}_i \in \mathbb{R}^N$ ③, where the value of $\hat{C}_i^k$ represents the probability that the predicted label $\hat{y}_i = k$. This correlation vector is then converted to a real-valued prediction by taking the expected value ④. Table 1 summarizes the notations used in this work.

Prior works use custom-designed label encoding in this formulation. For example, Shah et al. (2022) proposed a series of suitable label encodings $B_i = \mathcal{E}(Q_i) \in \{0, 1\}^M$. The network can be trained using binary cross-entropy loss between $B_i$ and $\hat{Z}_i$, and these encodings are used as columns of the decoding matrix ($D_{:,i} = \mathcal{E}(i)$). However, it is desirable to automatically find suitable encodings $B_i$ and decoding matrix $D$ without searching through a set of hand-designed encodings.

The search space of binary label encodings is discrete and hence challenging to search using traditional continuous optimization methods (Darabi et al., 2018). Hence, we relax the assumption of binarized label encodings and use a continuous search space. This relaxation, coupled with the proposed formulation, enables the use of traditional optimizers to learn label encoding and the decoding matrix $D$ with the entire network by optimizing the loss between targets and prediction. Let $\mathbb{S}_n$ represent the set of training samples with quantized target $Q_i = n$, and $E \in \mathbb{R}^{N \times M}$ represent a label encoding matrix, where each row $E_{n,:}$ is the encoding for target $Q_i = n$. $E$ is defined as:

$$E_{n,:} = \frac{1}{|\mathbb{S}_n|} \sum_{i \subset \mathbb{S}_n} \hat{Z}_i \qquad (1)$$

However, training the network solely with the loss between $\hat{y}_i$ and $y_i$ does not constrain the search space of label encodings ($E$). In regression, the label encoding ($E$) significantly impacts the accuracy, and label encodings that follow specific properties result in lower error (Shah et al., 2022). The following section explains desirable characteristics of output codes for regression and how these properties can be encouraged in learned label encoding using regularization functions.

## 3.2 LABEL ENCODING LEARNING WITH REGULARIZERS

Section 2.1 summarizes the properties of suitable binary label encodings for regression proposed by prior works. These properties constrain the vast search space of label encodings. We further propose two properties applicable to real-valued label encodings to narrow its search space.

**R1 - Distance between encodings:** A binary classifier's real-valued output represents its confidence (i.e., error probability). The L1 distance between real-valued predicted encodings gives more weight to classifiers that are more confident (i.e., higher value). Thus, by considering the L1 distance between real-valued codes instead of the hamming distance between binary codes, we can combine the second and third design properties of binary label encodings (Section 2.1) into a single rule for real-valued label encodings. This gives the first regularization rule: *L1 distance between encodings for two labels should increase with the difference between two labels*, i.e., $||E_{i,:} - E_{j,:}||_1 \propto |i - j|$.

**R2 - Regularizing bit transitions:** The number of bit transitions in a bit-position of label encoding gives a measure of the binary classifier's decision boundary's complexity. There are no $0 \rightarrow 1$ or $1 \rightarrow 0$ transitions in real-valued label encodings. Thus, we approximate the number of bit transitions by measuring the L1 distance between encodings for adjacent label values $Q_i = n$ and $Q_i = n + 1$. The number of bit transitions for real-valued label encoding $E$ can be approximated as:

$$\sum_{i=1}^{M} \sum_{j=1}^{N-1} |E_{j,i} - E_{j+1,i}| \tag{2}$$

This leads to the second regularization rule: *The L1 distance between encodings for adjacent target label values should be regularized to find a balance between the complexity of the decision boundary and the error-correction capability of designed codes for a given benchmark.*

### 3.3 Loss Function Formulation

We propose two regularizers applicable to learned label encoding ($E$) to limit its search space. $E$ is measured from the output codes $\hat{Z}_i$ over the complete training dataset (Equation 1). However, deep neural networks are trained using mini-batches, where each batch consists of $K$ training examples sampled randomly from the (typically shuffled) training set. We extend the proposed regularization rules to apply to a minibatch-based loss function.

**R1:** Regularizer R1 can be approximated as the following for a batch with $K$ training examples:

$$\mathcal{L}_1 = \sum_{i=1}^{K} \sum_{j=1}^{K} \max(0, 2 \times |y_i - y_j| - ||\hat{Z}_i - \hat{Z}_j||_1) \tag{3}$$

The above regularization considers $K^2$ pairs in a minibatch of $K$ samples, and penalizes a pair of training samples $i$ and $j$ if L1 distance between encodings $\hat{Z}_i$ and $\hat{Z}_j$ is less than twice the difference between corresponding label values. The scaling parameter is set to two as it encourages at least one bit difference between two binary codes. This encourages the L1 distance between encodings to be approximately proportional to the difference between corresponding target values.

**R2:** Regularizer R2 minimizes the L1 distance between encodings of adjacent label values. In a randomly formed minibatch consisting of only a subset of training examples, adjacent target labels might not be present. Hence it is nontrivial to apply this regularizer to the label encoding. However, we find that imposing this regularizer on the decoding matrix also helps with regularizing the bit transitions in the learned label encoding and can be added to the loss function irrespective of the batch formulation approach. Theoretical analysis and empirical verification for this approach are provided in Section A.1.7 and Section 4.4. Equation 4 represents the proposed regularizer.

$$\mathcal{L}_2 = \sum_{i=1}^{M} \sum_{j=1}^{N-1} |D_{i,j} - D_{i,j+1}| \tag{4}$$

**Loss Function:** We use the cross-entropy loss between $\hat{C}_i$ and soft target labels. Here, each bit of label encoding resembles a binary classifier. However, identifying the predicted label corresponding to the multi-bit label encoding can be treated as a multiclass classification problem. Soft target labels are probability distributions generated using the distance between different classes. Soft target labels can be used with cross-entropy loss and have shown improvement over typical classification loss between the correlation vector $\hat{C}_i$ and quantized target label $Q_i$ or regression loss between the expected prediction $\hat{y}_i$ and target label $y_i$ for ordinal regression (Díaz & Marathe, 2019). We use

Table 2: Benchmarks used for evaluation

| Task | Feature Extractor | Dataset | Benchmark | Label range/ Quantization levels | $\theta$ |
|---|---|---|---|---|---|
| Landmark-free 2D head pose estimation | ResNet50 (He et al., 2016) | 300LP (Zhu et al., 2016)/AFLW2000 (Zhu et al., 2016) | LFH1 | 0-200/200 | 10 |
| | | BIWI (Fanelli et al., 2013) | LFH2 | 0-150/150 | 10 |
| Facial Landmark Detection | HRNetV2-W18 (Wang et al., 2020) | COFW (Burgos-Artizzu et al., 2013) | FLD1/FLD1_s (100%/10% training dataset) | 0-256/256 | 10 |
| | | 300W (Sagonas et al., 2013) | FLD2/FLD2_s (100%/10% training dataset) | 0-256/256 | 10 |
| | | WFLW (Wu et al., 2018) | FLD3/FLD3_s (100%/10% training dataset) | 0-256/256 | 10 |
| Age estimation | ResNet50/ ResNet34 | MORPH-II (Ricanek & Tesafaye, 2006) | AE1 | 0-64/64 | 10 |
| | | AFAD (Niu et al., 2016) | AE2 | 0-32/32 | 10 |
| End-to-end autonomous driving | Pilot-Net(Bojarski et al., 2017) | PilotNet | PN | 0-670/670 | 10 |

this loss function for RLEL and multiclass classification. Complete loss function with regularizers (Equation 3 and 4) can be written as:

$$\mathcal{L} = \sum_{i=1}^{K} \text{CE}(\hat{C}_i, \phi(y_i)) + \alpha \sum_{i=1}^{M} \sum_{j=1}^{N-1} |D_{i,j} - D_{i,j+1}| + \beta \sum_{i=1}^{K} \sum_{j=1}^{K} \max(0, 2 \times |y_i - y_j| - ||\hat{Z}_i - \hat{Z}_j||_1),$$

$$\text{where } \phi^j(y_i) = \frac{e^{-|j-y_i|}}{\sum_{n=1}^{N} e^{-|n-y_i|}} \quad (5)$$

Here, the first term is the loss between target and predicted labels. $\phi_i$ represents the target probability distribution generated from target $y_i$. The second and third terms are for regularizer R1 (Equation 3) and regularizer R2 (Equation 4), respectively.

A trade-off exists between the proposed desirable properties of label encodings: Encouraging one design property comes at the cost of relaxing constraints imposed by other design properties. As demonstrated by Shah et al. (2022), finding the right balance between these properties for a given benchmark is crucial to finding the best label encoding for a given problem. Thus, these design properties can be naturally applied as regularizers, and the search for balance between different properties can be seen as tuning the regularization parameters $\alpha$ and $\beta$.

## 4 EVALUATION

This section first provides the experimental setup used to evaluate the proposed approach, then we compare RLEL with different label encoding design methods. We also compare RLEL with different regression approaches to demonstrate its effectiveness as a generic regression approach. Last, we provide an ablation study to show the impact of proposed regularizers.

### 4.1 EXPERIMENTAL SETUP

Table 2 summarizes the regression tasks, feature extractors architecture (Figure 3), and datasets for benchmarks used for evaluation. Selected benchmarks cover different tasks, datasets, and network architectures and have been used by prior works on regression due to the complexity of the task (Díaz & Marathe, 2019; Shah et al., 2022). We also evaluated RLEL on facial landmark detection tasks with smaller datasets to demonstrate its generalization capability. In this setup, a subset of training samples is used for training, whereas the complete test dataset is used to measure the test error.

Landmark-free 2D head pose estimation (LFH) takes a 2D image as input and directly finds the pose of a human head with three angles: yaw, pitch, and roll. The facial landmark detection task focuses on finding $(x, y)$ coordinates of key points in a face image. The age estimation task is used to find a person's age from the given face image. In end-to-end autonomous driving, the car's steering wheel's angle is to be predicted for a given image of the road. Normalized Mean Error (NME) or Mean Absolute Error (MAE) with respect to raw real-valued labels are used as the evaluation metrics.

We compare with other encoding design approaches, including simulated annealing, autoencoder (summarized in Appendix A.2), and manually designed codes (Shah et al., 2022). We also compare

Table 3: Comparison of RLELwith different label encoding design approaches. The bold and underlined numbers represent the first and second best errors, respectively.

| | Error (MAE or NME) | | | | | |
|---|---|---|---|---|---|---|
| Approach | LFH1 | LFH2 | FLD1 | FLD1_s | FLD2 | FLD2_s |
| Simulated annealing | 4.32±0.12 | 5.03±0.08 | 3.55±0.01 | 6.52±0.05 | 3.59±0.00 | 5.35±0.01 |
| Autoencoder | **3.38**±0.01 | 4.84±0.02 | 3.39±0.01 | 4.85±0.03 | 3.39±0.00 | 4.20±0.05 |
| LEL(w/o regularizers) | 4.03±0.15 | 4.96±0.08 | 3.36±0.01 | 4.98±0.07 | 3.39±0.01 | 4.28±0.05 |
| BEL(Manually designed) | 3.56±0.11 | **4.77**±0.05 | **3.34**±0.01 | **4.63**±0.03 | 3.40±0.02 | **4.15**±0.01 |
| RLEL | 3.55±0.10 | **4.77**±0.05 | 3.36±0.01 | 4.71±0.04 | **3.37**±0.02 | **4.15**±0.05 |

| Approach | FLD3 | FLD3_s | AE1 | AE2 | PN |
|---|---|---|---|---|---|
| Simulated annealing | 4.52±0.02 | 6.38±0.01 | 2.33±0.01 | 3.17±0.01 | 4.25±0.01 |
| Autoencoder | 4.36±0.01 | 5.62±0.01 | 2.29±0.00 | 3.19±0.01 | 4.49±0.04 |
| LEL(w/o regularizers) | **4.35**±0.02 | 5.68±0.04 | 2.30±0.01 | 3.17±0.01 | 3.22±0.02 |
| BEL(Manually designed) | 4.36±0.02 | 5.62±0.00 | **2.27**±0.01 | **3.11**±0.00 | 3.11±0.01 |
| RLEL | **4.35**±0.01 | **5.58**±0.01 | 2.28±0.01 | 3.14±0.01 | **3.01**±0.03 |

RLEL with generic regression approaches, such as direct regression and multiclass classification. For direct regression, L1 or L2 loss functions with L2 regularization are used. Label value scaling (hyperparameter) is used to change the numeric range of labels. For multiclass classification, we use cross-entropy loss between the softmax output and target labels.

The feature extractor and regressor are trained end-to-end for all approaches. The feature extractor architecture, data augmentation, and the number of training iterations are kept uniform across different approaches for a given benchmark. There is no notable difference between the training time for all approaches. The training dataset is divided into 70% training and 30% validation sets for tuning hyperparameters. The network is trained using the full dataset after hyperparameter tuning. We use the same values for quantization levels as prior work (Shah et al., 2022). An average of five training runs with an error margin of 95% confidence interval is reported. Appendix A.3 provides details on datasets, training parameters, related work (task-specific approaches), and evaluation metrics.

## 4.2 COMPARISON OF RLEL WITH ENCODING DESIGN APPROACHES

Table 3 compares different encoding design approaches. RLEL results in lower error than simulated annealing and autoencoder-based approaches for most benchmarks. Both approaches are widely used for code design. However, for regression tasks, the suitability of label encoding depends upon the problem, including the task, network architecture, and dataset (Shah et al., 2022). Simulated annealing or autoencoder-based approaches do not optimize the encodings end-to-end with the regression problem., resulting in higher error. Furthermore, the gap between the error of learned label encoding with and without regularizers (RLEL and LEL) increases for smaller datasets, which suggests that RLEL-learned codes generalize better.

RLEL can not be used with binary-cross entropy loss for training. We observe that for some benchmarks, the autoencoder-based approach outperforms (e.g., LFH1) as it can be used with binary-cross entropy loss. The main objective of RLEL is to automatically learn label encoding that can reach the accuracies of manually designed codes (BEL), as using such codes is time and resource-consuming. Hyperparameter search for RLEL can be performed by off-the-shelf hyperparameter tuners/libraries without manual efforts (Li et al., 2017; Falkner et al., 2018). In contrast, hand-designed codes need human intervention to design codes. Also, multiple training runs are still required to find suitable codes for a given benchmark from a set of hand-designed codes. As shown in Table 3, *RLEL results in lower or comparable errors to hand-designed codes.*

## 4.3 COMPARISON OF RLEL WITH REGRESSION APPROACHES

RLEL is a generic regression approach that focuses on regression by binary classification and proposes a label encoding learning approach. We compare RLEL with other generic regression approaches, including direct regression and multiclass classification as shown in Table 4. RLEL problem formulation introduces more fully-connected layers after the feature extractor; hence, we also perform an ablation study on increasing the number of fully connected layers in Appendix A.1.4.

Table 4: Comparison of RLELwith different regression approaches and state-of-the-art task-specialized approaches (more details in Appendix A.3). "/$x$M" represents the model size.

|  | Direct regression | Multiclass classification | RLEL | Task-specialized approach* |
|---|---|---|---|---|
| LFH1 | 4.22±0.13/23.5M | 4.49±0.24/24.2M | 3.55±0.10/23.6M | 3.30±0.04/69.8M |
| LFH2 | 5.32±0.12/23.5M | 5.31±0.05/24.8M | 4.77±0.05/23.6M | 3.90±0.03/69.8M |
| FLD1 | 3.60±0.02/10.2M | 3.48±0.03/25.6M | 3.36±0.01/10.6M | 3.34±0.02/10.6M |
| FLD1_s | 32.70±1.37/10.2M | 5.36±0.03/25.6M | 4.71±0.04/10.6M | - |
| FLD2 | 3.54±0.03/10.2M | 3.46±0.02/45.2M | 3.37±0.02/11.2M | 3.07/25.1M |
| FLD2_s | 5.04±0.02/10.2M | 4.50±0.04/45.2M | 4.15±0.05/11.2M | - |
| FLD3 | 4.64±0.03/10.2M | 4.46±0.01/61.3M | 4.35±0.01/11.7M | 4.32/- |
| FLD3_s | 6.35±0.07/10.2M | 6.05±0.01/61.3M | 5.58±0.01/11.7M | - |
| AE1 | 2.37±0.01/23.5M | 2.75±0.03/24.2M | 2.28±0.01/23.6M | 1.96/3.7M |
| AE2 | 3.16±0.02/23.5M | 3.38±0.05/24.8M | 3.14±0.01/23.6M | 3.47/21.3M |
| PN | 4.24±0.45/10.2M | 5.54±0.03/25.6M | 3.01±0.03/10.6M | 4.24/10.2M |

*This uses different network architecture, data augmentation, and training process.

Table 5: Effect of regularization R2 on bit-transitions in binarized and real-valued label encodings.

| $\alpha$ value | Proposed regularizer using Decoding matrix (Equation 4) | #Bit transitions in binarized label encoding | Approximated bit transitions in label encoding (Equation 2) |
|---|---|---|---|
| 0 | 6816.1 | 5097 | 391.88 |
| 0.1 | 215.3 | 3596 | 168.52 |
| 0.5 | 130.8 | 3180 | 104.19 |

*RLEL consistently lowers the error compared to direct regression and multiclass classification with* $10.9\%$ *and* $12.4\%$ *improvement on average.*

## 4.4 ABLATION STUDY

Figure 1a demonstrates that the use of regularizer R1 encourages the L1 distance between encodings to be proportional to the difference between target values. The second regularizer R2 is introduced to regularize the number of bit transitions in encodings. As mentioned in Section 3.3, we apply the regularization on the decoding matrix as it is nontrivial to apply this regularization on the output codes for randomly formed batches. Table 5 summarizes the effect of $\alpha$ (i.e., the weight of R2) on the number of bit transitions in the decoding matrix and binarized/real-valued label encoding. The second column is the number of bit transitions in the decoding matrix (Equation 4), which is used as the regularization function. The third and fourth columns are the total number of bit transitions in binarized and real-valued encodings (Equation 2). The table shows that the proposed regularizer on the decoding matrix also encourages fewer bit transitions in the label encoding. Figure 1b shows the impact of regularizer R2 on learned binarized label encoding.

As pointed out by prior works (Shah et al., 2022), there is a trade-off between the error probability and error correction capability of classifiers for regression. Hence, depending upon the benchmarks, more bit transitions can be added as the advantage of increased error correction outweighs the increase in classification error. We observe a similar trend, where adding R2 does not improve error for some benchmarks (FLD1_s, FLD2_s), as it constrains the number of bit transitions.

## 5 CONCLUSION

This work proposes an end-to-end approach, Regularized Label Encoding Learning, to learn label encodings for regression by binary classification setup. We propose a combination of continuous approximation of binarized label encodings and regularization functions. This combination enables an efficient and automated search of suitable label encoding for a given benchmark using traditional continuous optimization approaches. The proposed regularization functions encourage label encoding learning with properties suitable for regression, and the learned label encodings generalize better, specifically for smaller datasets. Label encodings designed by the proposed approach outperform simulated annealing- and autoencoder-designed label encodings by $12.6\%$ and $2.1\%$, respectively. RLEL-designed codes show lower or comparable errors to hand-designed codes. RLEL reduces error on average by $10.9\%$ and $12.4\%$ over direct regression and multiclass classification.

## 6 ACKNOWLEDGEMENTS

This research has been funded in part by the National Sciences and Engineering Research Council of Canada (NSERC) through the NSERC strategic network on Computing Hardware for Emerging Intelligent Sensory Applications (COHESA) and through an NSERC Strategic Project Grant. Tor M. Aamodt serves as a consultant for Huawei Technologies Canada Co. Ltd and recently served as a consultant for Intel Corp.

**Reproducibility:** We have provided details on training hyperparameters, experimental setup, and network architectures in Appendix A.3. Code is available at `https://github.com/ubc-aamodt-group/RLEL_regression`. We have provided the training and inference code with trained models.

**Code of Ethics:** Autonomous robotics and vehicles are major applications of deep regression networks. Thus improvement of regression tasks can accelerate the progress of these fields, which may lead to some negative societal impacts such as loss of jobs, privacy, and ethical concerns.

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

# A  APPENDIX

This supplemental material provides additional results and ablation studies (Section A.1), methodology for baseline encodings design approaches (Section A.2), and related work on task-specialized approaches and experimental setup (Section A.3) for RLEL. Code is available at `https://github.com/ubc-aamodt-group/RLEL_regression`.

## A.1  ABLATION STUDY

Section A.1.1, Section A.1.2, and Section A.1.3 provide an ablation study and supporting data on impact of proposed regularization functions and hyperparameters on label encoding learning. Section A.1.4 covers an ablation study on the impact of the number of fully connected layers in direct regression and multiclass classification. Section A.1.5 explains and compares deep hashing approaches (adapted for regression) with RLEL. Section A.1.6 provides results for geometric mean and Pearson coefficient as evaluation metrics.

### A.1.1  IMPACT OF REGULARIZER R1

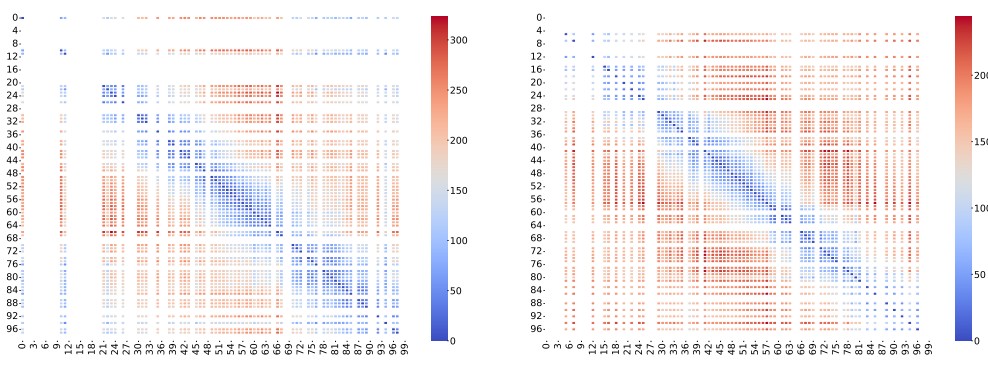

(a) Without regularization: $\beta = 0$     (b) With regularization: $\beta = 5.0$

Figure 4: (a) and (b) show the L1 distance between pairs of encodings for FLD1_s benchmark for $\beta = 0$ and $\beta = 5.0$, respectively. Each cell (i,j) in this matrix represents the L1 distance between learned encodings for label $i$ and $j$, i.e., $||E_{i,:} - E_{j,:}||_1$.

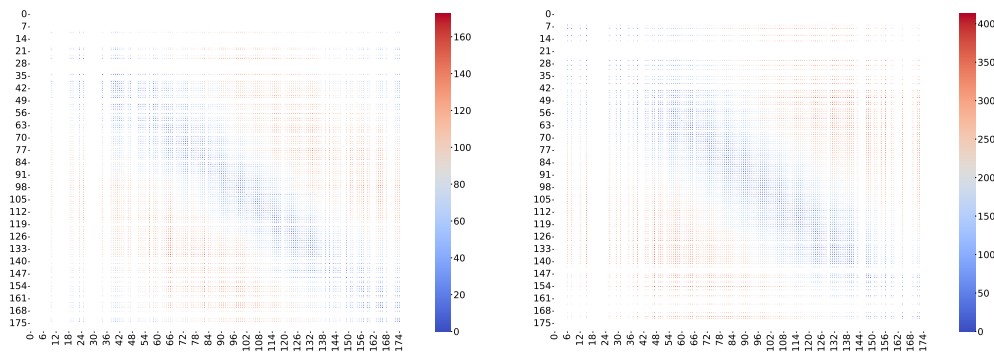

(a) Without regularization: $\beta = 0$     (b) With regularization: $\beta = 5.0$

Figure 5: (a) and (b) show the L1 distance between pairs of encodings for FLD2_s benchmark for $\beta = 0$ and $\beta = 5.0$, respectively. Each cell (i,j) in this matrix represents the L1 distance between learned encodings for label $i$ and $j$, i.e., $||E_{i,:} - E_{j,:}||_1$.

We proposed regularization function R1 to encourage the L1 distance between encodings to be proportional to the difference between corresponding label values. Figure 4a and Figure 4b represent the L1 distance between pairs of learned encodings for FLD1_s benchmark without and with regularization, respectively. The $X$-axis and $Y$-axis represent the label values. Here, some columns and rows are replaced by white lines, as these label values are not present in the training dataset. The data point at coordinates $(i, j)$ represent the L1 distance between encodings for label $i$ and $j$, i.e.,

Table 6: Effect of the scaling parameter on error for FLD1$_s$ and FLD2$_s$ benchmarks.

| Value of the scaling parameter | NME (FLD1$_s$) | NME (FLD2$_s$) |
|---|---|---|
| 1 | 4.89 | 4.15 |
| 2 | 4.71 | 4.15 |
| 3 | 4.83 | 4.20 |
| 4 | 4.97 | 4.27 |
| 5 | 4.95 | 4.28 |
| 6 | 5.06 | 4.41 |

$||E_{i,:} - E_{j,:}||_1$. For example, in Figure 4a, the L1 distance between encodings for label values 0 and 97 is $\sim 120$ (light-blue coloured point at coordinate $(0, 97)$). In Figure 4b, the L1 distance between encodings for label values 4 and 96 is $\sim 170$ (red coloured point at coordinate $(4, 96)$).

The first design property (Section 3) states that the L1 distance between encodings should increase with the difference between corresponding label values. The difference between label values for pairs of encodings increases with the distance from the diagonal of this plot. Thus, the value of data points (i.e., the L1 distance between encodings) should increase with the distance from the diagonal of this plot. As shown in Figure 4a, without regularization, the distance between encodings is less for faraway label values (blue-colored data points away from diagonal), which shows that learned encodings do not follow the proposed design property. As shown in Figure 4b, the introduction of regularization function R2 remedies this and increases the L1 distance between encodings for faraway labels. Similar observations are made for FLD2_s benchmarks, as shown in Figure 5a and Figure 5b.

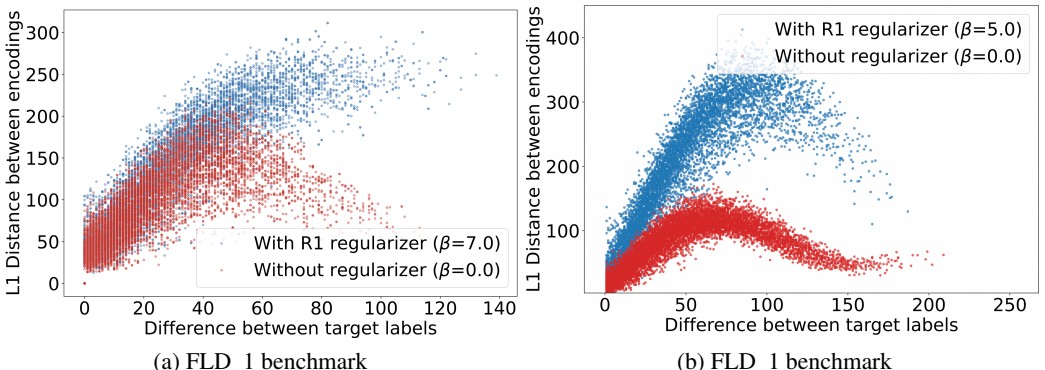

(a) FLD_1 benchmark          (b) FLD_1 benchmark

Figure 6: (a) and (b) plot the L1 distance between pairs of encodings versus distance between corresponding label values for FLD1_s and FLD2_s benchmarks.

Figure 6 plots the L1 distance between encodings versus the difference between corresponding label values for benchmarks FLD1_s and FLD2_s. For both the benchmarks, the proposed regularizer R1 helps enforce the first design property for real-valued label encodings and results in better label encodings with lower error (Table 3).

**Effect of the scaling parameter in equation 3**    We use the scaling parameter 2 in equation 3. Our intuition behind using the scaling parameter 2 is based on binary-encoded labels. For two adjacent labels (i.e., $|y_i - y_j| = 1$), the loss function encourages $||\hat{Z}_i - \hat{Z}_j||_1$ to be greater than 2. Here, $\hat{Z}$ is the output encodings. In the case of binarized label encoding ($-1$ if $Z < 0$ and $+1$ if $Z > 0$), $||Z_i - Z_j||_1 = 2$ signifies that two encodings differ in at least one bit.

We also analyzed the effect of changing this parameter for two benchmarks. Table 6 shows the impact of changing this scaling parameter for two benchmarks. We observe that the error is higher if the scaling parameter is too low, as encodings for two adjacent labels can not be discriminated against. If this parameter is set too high, the encoding space is more constrained and consequently the performance is degraded.

Based on this intuition and empirical verification on two benchmarks, we use the value 2 for all benchmarks.

### A.1.2 IMPACT OF REGULARIZER R2

The regularization function R2 is proposed to reduce the number of bit transitions in the learned label encoding. Figure 7 compares the label encodings learned for LFH1 benchmark for different values of $\alpha$, where $\alpha$ is the weight of regularization function R2 (Equation 5). Each row $k$ is the encoding for label value $k$. Each column $k$ represents the output of the encoding position $k$ for different label values. The regularization function is proposed to decrease the transitions in an encoding bit (blue→red and red→blue) over the range of label values. Section 4.4 provided quantitative results to demonstrate that increasing the value of $\alpha$ reduces the number of bit transitions. We observe similar trends in the plots of learned label encodings shown in Figure 7; increasing the value of $\alpha$ decreases bit transitions in the learned label encodings and improves MAE.

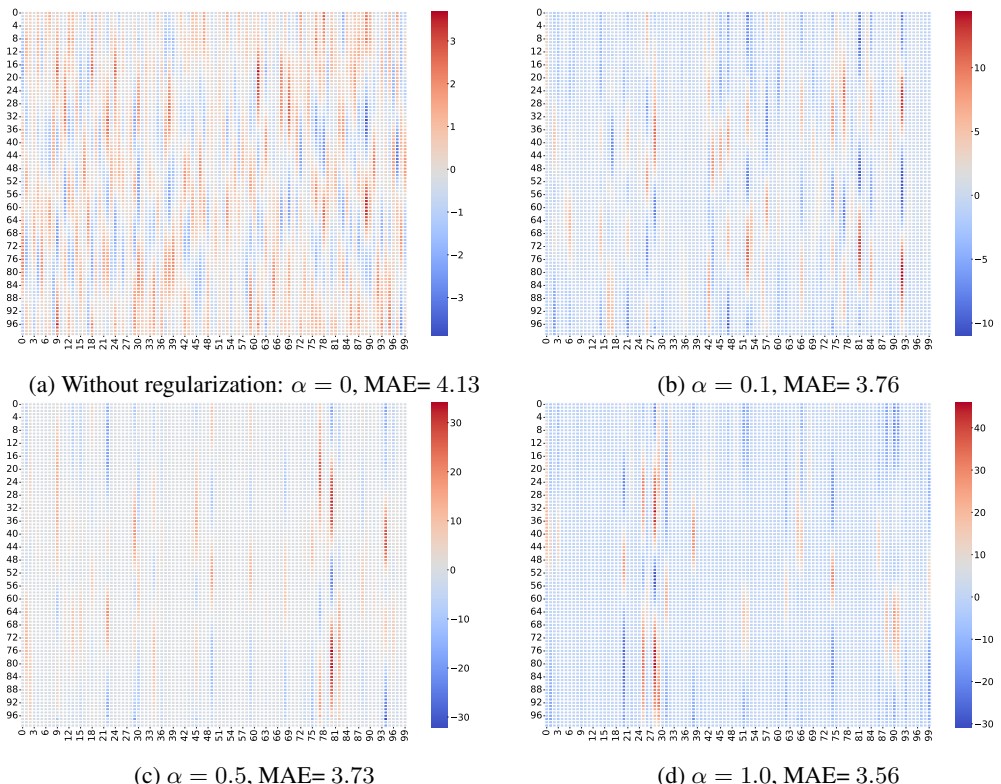

(a) Without regularization: $\alpha = 0$, MAE= 4.13       (b) $\alpha = 0.1$, MAE= 3.76

(c) $\alpha = 0.5$, MAE= 3.73      (d) $\alpha = 1.0$, MAE= 3.56

Figure 7: (a)-(d) represent the label encodings learned by RLEL for different values of weight $\alpha$ for regularizer R2 (Equation 5).

### A.1.3 EFFECT OF HYPERPARAMETERS IN RLEL

The RLEL approach introduces two hyperparameters. We first evaluate the sensitivity to these hyperparameters to determine the complexity of hyperparameter tuning. Figure 8 shows the NME for FLD1_s benchmark for different values of $\alpha$ and $\beta$ values in Equation 5. As shown in the figure, the error is not sensitive to small changes in these hyperparameters' values, suggesting that a sparse search in the hyperparameter space suffices. Furthermore, several approaches have been proposed for efficient hyperparameter search (Li et al., 2017; Falkner et al., 2018), and any off-the-shelf hyperparameter tuners/libraries can be used to automatically find these values without manual efforts. In contrast, hand-designed codes need human intervention to design codes. Also, multiple training runs are still required to find suitable codes for a given benchmark from a set of hand-designed codes. On the other hand, RLEL provides an end-to-end automated approach for label encoding learning.

### A.1.4 IMPACT OF THE NUMBER OF FULLY-CONNECTED LAYERS:

For RLEL , we use an extra fully connected bottleneck layer in the regressor as proposed by the prior work on regression by binary classification (Shah et al., 2022). We provide an ablation study

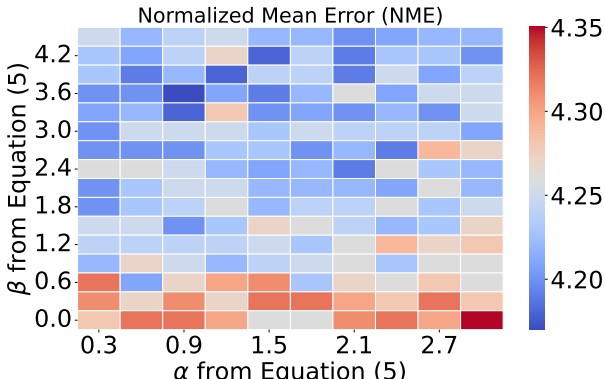

Figure 8: Impact of hyperparameters $\alpha$ and $\beta$ from Equation 5 on NME for FLD1_s benchmark.

Table 7: Impact of the number of fully-connected layers in direct regression and multiclass classification on the error (MAE or NME). This table is reproduced from (Shah et al., 2022).

| Benchmark | Direct regression | | Multiclass classification | |
|---|---|---|---|---|
| | 1 FC layer | 2 FC layers | 1 FC layer | 2 FC layers |
| LFH1 | 4.76 | 5.19 | 4.49 | 4.82 |
| LFH2 | 5.65 | 5.59 | 5.31 | 5.42 |
| FLD1 | 3.60 | 3.63 | 3.58 | 3.56 |
| FLD2 | 3.54 | 3.58 | 3.51 | 3.62 |
| FLD3 | 4.64 | 4.63 | 4.50 | 4.64 |
| FLD4 | 1.51 | 1.51 | 1.56 | 1.53 |
| AE1 | 2.44 | 2.35 | 2.75 | 2.81 |
| AE2 | 3.21 | 3.14 | 3.38 | 3.40 |
| PN | 4.24 | 4.33 | 4.56 | 5.74 |

(reproduced from (Shah et al., 2022)) to show the impact of additional fully connected layers in direct regression and multiclass classification. Table 7 provides the error (MAE or NME) for direct regression and multiclass classification with one or two fully connected layers after the feature extractor. As shown in the table, increasing the number of fully connected layers in direct regression and multiclass classification does not reduce the error for most benchmarks (possibly due to overparameterization).

### A.1.5    COMPARISON WITH DEEP HASHING APPROACHES

Deep supervised hashing approaches use neural networks as a hash function and learn hash codes in an end-to-end manner. The loss function for deep supervised hashing is designed to preserve the similarity between inputs in the hashing space. Often, these approaches use the label information to determine the similarity between images (i.e., same label) (Liu et al., 2016; Xia et al., 2014). Some deep hashing approaches have proposed to augment the loss function with classification loss to improve the performance. We adapt two widely used deep-hashing approaches to regression and compare RLEL with deep hashing approaches.

Liu et al. (2016) proposed a deep supervised hashing (DSH) approach with a loss function based on the pairwise similarity between images. The proposed approach introduces a loss function to preserve the similarity between output codes for similar training images (e.g., same class) and maximize discriminability between output codes for different training images (e.g., different class). Further, they propose using relaxation on the binary output and a regularizer to encourage the output code to be close to discrete values $+1/-1$. The hamming distance between output codes can be computed for binary-like outputs using the L2 norm. We use DSH for regression with some modifications (DSH-reg). We used the quantized label to determine the class of a training sample.

Lai et al. (2015) proposed a triplet ranking loss to learn a hash function that preserves relative similarities between images (SFLH). For images $(I, I+, I-)$, where $I$ is closer to $I+$ than $I-$, the loss function is designed to encourage higher hamming distance between codes for $(I, I-)$ than $(I, I+)$. For classification datasets, triplets are typically formed using two images from the same class

and one from a different class (Norouzi et al., 2012). They proposed to use a piece-wise threshold function to encourage binary-like outputs.

We use the above approach (SFLH) for regression with a few modifications (SFLH-reg). To generate triplets, we pick sets of three images from a given batch and determine the similarity between images using differences between the label values. We use $K^2$ triplets for a minibatch of $K$ training samples.

Further, for both DSH-reg and SFLH-reg, we augment the loss function with regression loss. We add a fully-connected layer between the output code and prediction. The MSE loss between the final outputs and target labels is added to the loss function (DSH-reg-L2, SFLH-reg-L2).

Table 8: Comparison of RLELwith different deep hashing approaches adapted for regression.

| Method | MAE |
|---|---|
| DSH-reg | 71.3 |
| DSH-reg-L2 | 4.11 |
| SFLH-reg | 69.8 |
| SFLH-reg-L2 | 4.73 |
| RLEL ( only R1 ) | 3.93 |
| RLEL ( R1 + R2 ) | 3.55 |

Table 8 compares the modified deep hashing approaches with RLEL. The gap between loss functions with and without regression loss is significant, which shows that a loss function that only aims to preserve the similarity between output codes is not sufficient and needs to account for the error between decoded output and target (i.e., regression loss). RLEL results in a lower error as it is designed for regression problems that account for classifiers' nonuniform error probability distribution.

Regularizer R1 encourages the distance between output codes for images to be proportional to the difference between label values, similar to pairwise or ranking-based loss functions proposed by deep hashing. However, deep hashing approaches use the hamming distance between binary outputs. As we show in Section 3.2, the hamming distance between codes does not account for the error probability of classifiers. Thus we use the L1 distance between the real-valued outputs to account for the confidence of the classifiers. R1 does not use regularizer or nonlinear activation on the output codes to encourage binary-like outputs, as typically done in deep hashing approaches. In contrast, we show that suitable regression codes can be learned by not using this constraint. Thus RLEL with only R1 regularizer results in lower error than deep hashing approaches.

### A.1.6 EVALUATION

Table 9: Comparison of RLELwith different regression approaches using Geometric mean and Pearson coefficient as evaluation metrics.

| | RLEL | | Direct Regression | | Multiclass Classification | |
|---|---|---|---|---|---|---|
| | GeoMean | Pearson Coeff. | GeoMean | Pearson Coeff. | GeoMean | Pearson Coeff. |
| LFH1 | 1.95 | 97.68 | 2.91 | 97.10 | 2.30 | 94.60 |
| LFH2 | 2.09 | 92.22 | 2.49 | 91.06 | 2.40 | 88.76 |
| FLD1 | 0.96 | 99.94 | 1.07 | 99.93 | 1.04 | 99.93 |
| FLD1_s | 1.31 | 99.87 | 6.38 | 99.81 | 1.81 | 99.80 |
| FLD2 | 1.92 | 99.97 | 2.12 | 99.97 | 2.07 | 99.97 |
| FLD2_s | 2.44 | 99.96 | 3.03 | 99.98 | 3.22 | 99.94 |
| FLD3 | 0.96 | 99.99 | 1.05 | 99.99 | 1.01 | 99.97 |
| FLD3_s | 1.21 | 99.98 | 1.56 | 99.97 | 1.37 | 99.97 |

Table 9 compares RLEL with direct regression and multiclass classification using geometric mean and Pearson coefficient as evaluation metrics. The geometric mean represents the geometric mean of absolute error for the test dataset. The Pearson coefficient represents the correlation between the target and predicted labels for the test dataset. As shown in the table, RLEL results in significant reduction in the error compared to other generic regression approaches.

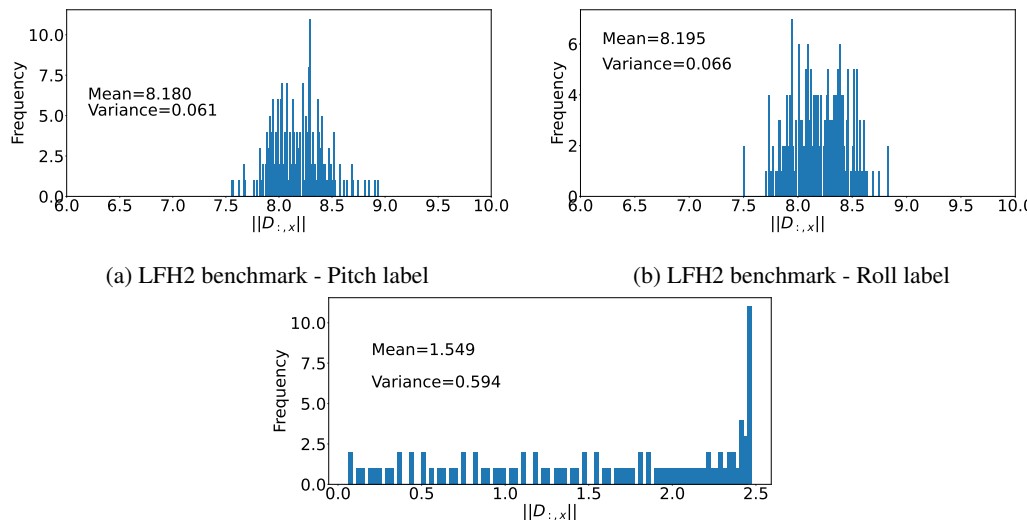

(a) LFH2 benchmark - Pitch label        (b) LFH2 benchmark - Roll label

(c) LFH1 benchmark - Roll label

Figure 9: (a) and (b) plot the distribution of $||D_{:,x}||$ for LFH2 benchmark. (c) plots the distribution of $||D_{:,x}||$ for LFH1 benchmark. Here the variance is very low, which suggests that the assumption $||D_{:,x}|| \approx ||D_{:y}||, x \in [1, N], y \in [1, N]$ is valid. For the LFH1 benchmark, the variance is higher than LFH2. However, all outliers are for label values with very few (or even zero) training examples.

### A.1.7   Theoretical Analysis of Proposed Regularization Functions

Regularization function R2:

We used matrix $D$ instead of label encoding $E$ to apply regularizer R2 in equation 4. We insight into this decision as follows. First, note the output encodings are multiplied with $D$ to generate the correlation vector $\hat{C}_i$ (Figure 3). We use the multiclass classification loss between $\hat{C}_i$ and the target labels for training. Due to this, label encoding $E$ and decoding matrix $D$ are related, and use of matrix $D$ proves to be effective for regularizer R2. We further explain this in detail below.

Let $E$ represent an encoding matrix of size $N \times M$. Each row $E_{k:}$ represents the encoding output when the label is $k$. $D$ is the decoding matrix of size $M \times N$. Let $\hat{C}_k$ represent the output correlation row vector of size $1 \times N$ when the target label is $k$. Here, $\hat{C}_k$ is obtained by multiplying $E_{k,:}$ with $D$ (Figure 3).

$$\hat{C}_k = E_{k,:}D \tag{6}$$

Since we apply softmax on the output vector to find the predicted label (Figure 3), ideally, $\hat{C}_k^k$ should have the highest value as the target label value is $k$.

$\therefore \hat{C}_k^k > \hat{C}_k^x$, where, $x \neq k, x \in \{1, 2, ..., N\}$

$\therefore E_{k,:}.D_{:,k} > E_{k,:}.D_{:,x}$, where, $x \neq k, x \in \{1, 2, ..., N\}$ (Using equation 6). Let $\theta_{k,x}$ represent the angle between row vector $E_{k,:}$ and column vector $D_{:,x}$. This leads to the below equation:

$$||E_{k,:}||||D_{:,k}||\cos(\theta_{k,k}) > ||E_{k,:}||||D_{:,x}||\cos(\theta_{k,x}), \text{ where, } x \neq k, x \in \{1, 2, ..., N\} \tag{7}$$

Shah et al. (2022) used a hand-crafted decoding matrix $D$ with an equal number of 1s and 0s in each column for binary-encoded labels. Hence the L2 norm of each column is the same. In label encoding learning, parameters of matrix $D$ are learned during training and are not constrained to have the same L2 norm for each column. However, we observe a similar trend empirically. Figure 9c plots the distribution of $||D_{:,x}||$ for different benchmarks. As shown in the figure, for most benchmarks, we observe a small variance in the distribution of $||D_{:,x}||$. Based on this intuition and empirical

validation, we assume that $||D_{:,x}|| \approx ||D_{:,y}||$ for $x \in [1, N]$ and $y \in [1, N]$ to simplify the analysis.

Using this assumption in equation 7 leads to the following inequality:

$\cos(\theta_{k,k}) > \cos(\theta_{k,x})$, where, $x \neq k, x \in \{1, 2, ..., N\}$

Thus the cosine similarity between $E_{k,:}$ and $D_{:,k}$ should be the highest to predict the label $k$. The optimization process to reduce the loss between the target and prediction will try to maximize this cosine similarity. In the best case, the angle between $E_{k,:}$ and $D_{:,k}$ will be zero, and both vectors are parallel.

This simplification leads to the following relation between $E$ and $D$.

$E_{k,:} = tD_{:,k}$, where $t > 0$

Similarly, $E_{k+1,:} = t'D_{:,k+1}$, where $t' > 0$

Since $t$ and $t'$ both are positive values, reducing $D_{i,k} - D_{i,k+1}$ also reduces $E_{k,i} - E_{k+1,i}$.

Regularizer rule R2 proposes to regularize the number of decision boundaries by regularizing $\sum_{i=1}^{M} \sum_{j=1}^{N-1} |E_{j,i} - E_{j+1,i}|$ as per equation 2. Based on the analysis above, regularizing $\sum_{i=1}^{M} \sum_{j=1}^{N-1} |D_{i,j} - D_{i,j+1}|$ helps with the above goal as $E_{j,i} - E_{j+1,i}$ reduces with $D_{i,j} - D_{i,j+1}$.

Regularization function R1:

The first property suggests $||E_{i,:} - E_{j,:}||_1 \propto |i - j|$.

So ideally, $||E_{i,:} - E_{j,:}||_1 = \lambda |i - j|$

Since $E_{x,:}$ is average of $\hat{Z}_i$ for samples with label value $x$ (equation 1), the above condition leads to:

$$||\hat{Z}_i - \hat{Z}_j||_1 = \lambda |y_i - y_j| \tag{8}$$

Based on this requirement, we add a regularization function $\max(0, \lambda |y_i - y_j| - ||\hat{Z}_i - \hat{Z}_j||_1)$, which penalizes the encodings if $||\hat{Z}_i - \hat{Z}_j||_1 < \lambda |y_i - y_j|$. It does not strictly impose equation 8. However, it approximately imposes the constraint as per shown in empirical verification in Section A.1.1.

Our intuition behind using the scaling parameter 2 is based on binary-encoded labels. For two adjacent labels (i.e., $|y_i - y_j| = 1$), the loss function encourages $||\hat{Z}_i - \hat{Z}_j||_1$ to be greater than 2. Here, $\hat{Z}$ is the output encodings. In the case of binarized label encoding ($-1$ if $Z < 0$ and $+1$ if $Z > 0$), $||Z_i - Z_j||_1 = 2$ signifies that two encodings differ in at least one bit.

### A.1.8 IMPACT OF THE NUMBER OF QUANTIZATION LEVELS ($N$)

The number of quantization buckets is treated as a design parameter for binary-encoded labels. Shah et al. (2022) showed that the error changes with the number of quantization levels. Fewer levels introduce quantization error, and more levels increase the number of bits in the encoding. They showed a trade-off between these two factors to decide the number of quantization levels.

Our work focuses on the design space of encoding and decoding functions. Hence we use the same values for the quantization levels ($N$) as BEL Shah et al. (2022). Parameter $N$ tuning can be integrated into hyperparameter tuning or included in the optimization process.

We further analyze the effect of the number of quantization levels for RLEL. Table 10 shows the NME (Normalized Mean Error) for different values of $N$ for FLD1 benchmark.

This suggests that the proposed method RLEL is less sensitive to the number of quantization levels for higher values. For RLEL, the decoding matrix that converts the encodings to the predicted label

Table 10: Impact of the number of quantization levels on error for FLD1 benchmark

| Quantization levels (N) | NME |
|---|---|
| 32 | 3.49 |
| 64 | 3.36 |
| 128 | 3.36 |
| 256 | 3.36 |
| 384 | 3.37 |
| 512 | 3.37 |

Table 11: Effect of dataset size on the error for FLD1 benchmark.

| %Dataset used | RLEL | BEL | Difference (RLEL-BEL) |
|---|---|---|---|
| 100 | 3.36 | 3.35 | 0.01 |
| 80 | 3.43 | 3.42 | 0.01 |
| 60 | 3.53 | 3.47 | 0.06 |
| 40 | 3.77 | 3.72 | 0.05 |
| 20 | 4.08 | 4.04 | 0.04 |
| 10 | 4.71 | 4.63 | 0.08 |

is also learned during the training (Figure 3). This matrix is of size $M \times N$, where each column represents the weight parameters for one quantization level. One possible reason for the above results is that matrix $D$ learns the number of quantization levels suitable for this problem.

There is a potential to learn the number of quantization levels and non-uniform quantization using the proposed RLEL framework. For example, in Figure 3- step (4), fixed parameters $j$ are used to scale the correlation vector $\hat{C}_i^j$ and find the expected prediction $\hat{y}_i$. These parameters represent quantization levels. One possible approach to learning the quantization levels is to make these parameters trainable. In this case, L1/L2 loss between the expected prediction $\hat{y}_i$ and target labels $y_i$ can be used to train the network.

### A.1.9 IMPACT OF DATASET SIZE ON ERROR FOR RLEL AND BEL

In order to compare the effect of dataset size on encoding design, we run BEL and RLEL approaches with the same training loss function (cross entropy loss in equation 5). We take the dataset FLD1 and use a fraction of the dataset for training. The entire test dataset is used for testing here. Table 11 summarizes the error achieved by RLEL and BEL for different fractions of the training dataset. The evaluation shows that the gap between the performance of RLEL and BEL decreases with the increase in dataset size, which suggests that RLEL might be able to achieve lower error for larger datasets.

### A.1.10 COMPARISON OF LEARNED AND MANUALLY DESIGNED ENCODINGS

We visually compare the encoding learned by RLEL with BEL manually designed code for one benchmark. Figure 10 shows the learned and manually designed encodings. Here, row $k$ represents an encoding for label $k$. Column $j$ represents the bit values for classifier-k over the numeric range of labels. We notice some common characteristics between both encodings. For example, the codes for nearby labels differ by fewer bits than faraway labels. Both the codes also have fewer bit transitions ($0 \rightarrow 1$ and $1 \rightarrow 0$ transitions in a column). These characteristics in the learned encodings are encouraged by the proposed regularizers R1 and R2. There are a few differences between learned and hand-crafted encodings. In contrast to hand-crafted labels, encodings for adjacent labels do not differ in some cases, where hand-crafted encoding assures at least one or two bits of difference between adjacent labels.

### A.2 LABEL ENCODING DESIGN

We evaluate different label encoding design approaches, including simulated annealing and autoencoder. These approaches have been used to design encodings for multiclass classification by prior works (Song et al., 2021; Cissé et al., 2012). We adapt these approaches to design encodings for

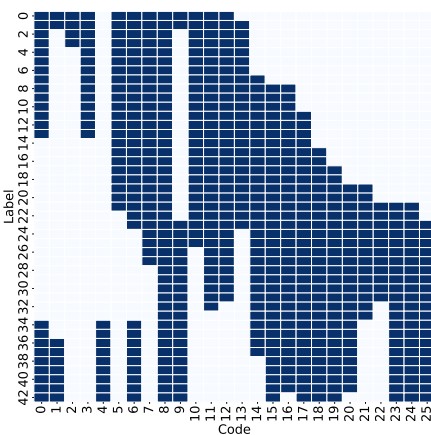 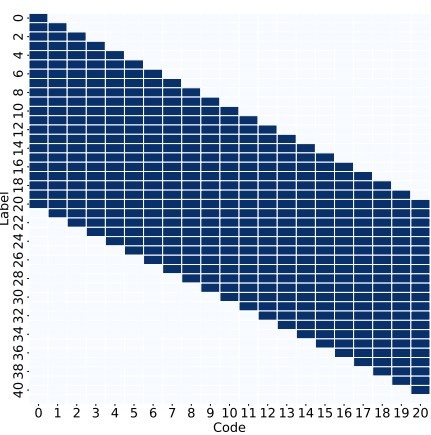

(a) Learned encoding (RLEL)        (b) Hand-crafted encoding (BEL)

Figure 10: (a) and (b) give examples of learned and hand-crafted encodings. Here, row $k$ represents an encoding for label $k$. Column $j$ represents the bit values for classifier-k over the numeric range of labels.

regression tasks and compare RLEL with these code design techniques. This section provides the methodology for simulated annealing and autoencoder-based label encoding design.

### A.2.1 SIMULATED ANNEALING

Simulated annealing is a probabilistic approach to find a global optimum of a given function. It is often used for combinatorial optimization, where the search space is discrete. Algorithm 1 represents

---

**Algorithm 1** Simulated annealing for encodings design

**Input:** $K_{max}$, T, $M$, $N$;
**Output:** C $\in \{0,1\}^{M \times N}$;

1: C = C0 $\in \{0,1\}^{N \times M}$, where $\Pr(C0_{i,j} = 0) = \Pr(C0_{i,j} = 1)$
2: t = T
3: **for** k $\in K_{max}$ **do**
4:   $C_{new}$ = Move(C)
5:   D = E($C_{new}$) - E(C)
6:   **if** D < 0 or $e^{\frac{-D}{t}}$ > Random(0,1) **then**
7:     C = $C_{new}$
8:   **end if**
9:   t = T / (k + 1)
10: **end for**

---

the pseudo-code for label encoding design using simulated annealing. This algorithm takes two hyperparameters, $K_{max}$ (number of iterations) and T (initial temperature). It designs a code matrix C of size $N \times M$, where $N$ is the number of values and $M$ is the number of bits. Each row $k$ in this code matrix represents encoding for value $k$. Code matrix C is initialized with a random matrix of 0 and 1 (Line 1).

For each iteration, a new code matrix $C_{new}$ is sampled from the current code matrix C using a Move function (Line 4). For example, a move function can be designed to randomly flip a few bits in C. The difference between *the errors* of the current and new code matrix is measured (Line 5). The error of a code matrix, i.e., expected regression error for this problem, is measured using function E. For example, E can be replaced by training a regression network for a given code matrix to measure the regression error. Finally, the current code matrix C is updated with the new matrix $C_{new}$

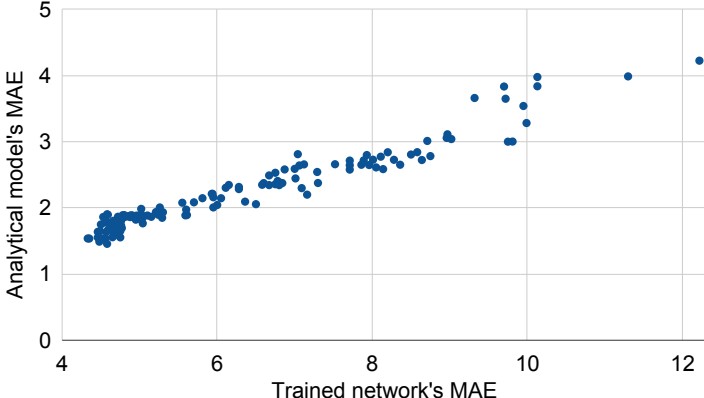

Figure 11: Comparison of Mean Absolute Error (MAE) approximated by the proposed analytical model and trained network for different code matrices.Each point in this plot is a different code matrix.

probabilistically. The probability is determined using the decrease in regression error and current temperature `t` (Line 6-8). The current temperature is updated for each iteration (Line 9).

There are mainly two design parameters in the above algorithm: the error measurement function `E` and the move function `Move`. We further explain the design of these functions.

**Error measurement:**

We used the expected absolute error between targets and decoded predictions for a given code matrix as its error, as the goal is to design a code matrix that results in lowest regression error. However, training a regression network for each sample code matrix to measure its regression error is computationally expensive and time-consuming ($\sim 200$ training runs). Hence we use an analytical model to estimate the regression error for a given code matrix.

Regression error is the absolute error between targets $Q_i$ and decoded predictions $\hat{Q}_i$. For a given target $Q_i$ and target code $B_i = C_{Q_i,:}$, the predicted code ($\hat{B}_i$) will be erroneous due to classification errors. This erroneous predicted code is decoded to a predicted value ($\hat{Q}_i$). The following equation is used to predict $\hat{Q}_i$ in expected-correlation-based decoding (Shah et al., 2022).

$$\mathcal{D}^{\text{GEN-EX}}(\hat{B}_i, C) = \sum_{k=1}^{N} k\sigma_k, \text{where } \sigma_k = \frac{e^{\hat{B}_i \cdot C_{k,:}}}{\sum_{j=1}^{N} e^{\hat{B}_i \cdot C_{j,:}}} \qquad (9)$$

The regression error can be estimated given sufficient samples of $B_i$ and $\hat{B}_i$.

Shah et al. (2022) provided an approximate model of classification errors. They showed that for each classifier, its error probability distribution can be approximated using a combination of $p$ Gaussian distributions, where $p$ is the number of bit transitions. Each Gaussian distribution is centered around a bit transition. For example, for bit-$k$ in unary code with bit transition between $Q = k$ and $Q = k + 1$, the error probability of the classifier-$k$ for different target labels $Q_i$ can be approximated as:

$$e_k(Q_i) = r f_{\mathcal{N}(\mu_k, \sigma^2)}(Q_i), \text{where, } \mu_k = k + 0.5 \qquad (10)$$

$\hat{B}_i$ can be sampled for the given $Q_i$ and $C$ using the above error-probability model. Equation 9 is then used to find the decoded prediction $\hat{Q}_i$. We measure the expected absolute error between $\hat{Q}_i$ and $Q_i$ using $100 \times N$ samples.

We further verify the validity of this analytical model by finding the correlation between regression error measured by this model and trained networks. Figure 11 plots the analytical regression error versus actual regression error for FLD_1 benchmarks. Here, each point is for a different code matrix. The $Y$-axis represents the absolute error approximated by the proposed analytical model. The $X$-axis

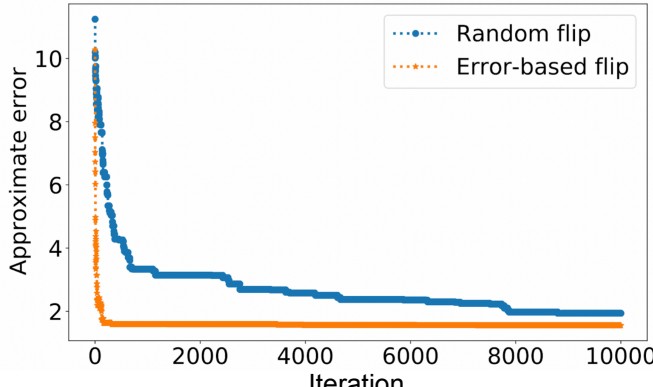

Figure 12: Comparison of convergence of random-flip and proposed error-based flip move functions.

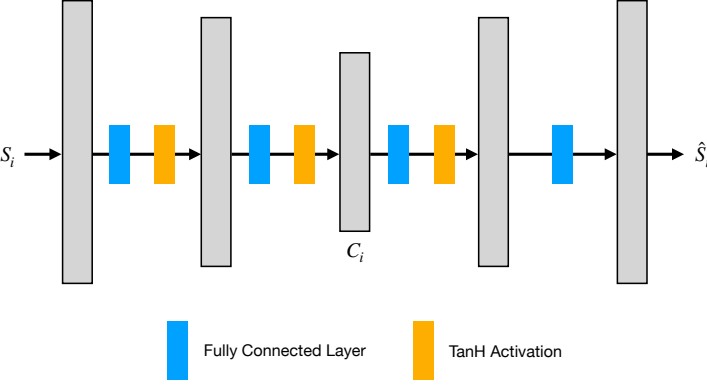

Figure 13: Network architecture for autoencoder-based encodings design.

represents the absolute test error of a trained network for a given code matrix. The figure shows that the proposed analytical model for error measurement approximates error with significant speedup.

**Move function:** The move function flips some bits in the current code matrix to sample a new one. A naive approach would be to randomly flip $b$ bits. We further optimize the move function to consider the regression task objective. For a given code matrix, using the proposed analytical model, we find a matrix $F$ of size $N \times N$, where $F_{i,j} = |i - j| \times Pr(\text{Round}(\mathcal{D}(\hat{B}, C)) = j | Q = i, B = C_{i,:})$. Thus, each element represents a pair $(C_{i,:}, C_{j,:})$ of encodings' contribution to expected error. We select top-$b$ pairs from this matrix. For each pair of encodings, we find bit-positions that have equal bit-values between two encodings, and a randomly selected bit-position from this list is flipped in encoding $C_{i,:}$. This procedure increases the hamming distance between pairs of encodings that contribute the highest to the regression error. Figure 12 compares the convergence of the proposed move function and a random-flip-based move function. Here the $Y$-axis represents the approximated error for the current code matrix, and $X$-axis represents the iteration identifier. The figure shows that the proposed error-based move function results in faster convergence and lower error.

We use the proposed move function with the analytical model to approximate regression error in Algorithm 1 to design label encoding for regression using simulated annealing.

### A.2.2 AUTOENCODER

Cissé et al. (Cissé et al., 2012) proposed an autoencoder-based approach to design encodings for a multiclass classification problem. Figure 13 represents the network architecture used for encodings design. Input $S_i$ is an $N$-dimensional vector for class $i$. Here, each element $S_i[j]$ represents the similarity between class $i$ and $j$. The output of the bottleneck layer $C_i$ is the designed encodings for class $i$.

For regression problems, we set $S_i[j] = |i - j|$. Let $W$ represent the weight parameters of the network. The network is trained using SGD optimization, where each batch consists of randomly sampled $i$ and $j$. The following loss function is used for training:

$$\mathcal{L} = ||\hat{S}_i - S_i||^2 + ||\hat{S}_j - S_j||^2 + \beta\max(0, b - ||C_i - C_j||_1) + \gamma||W||^2 \tag{11}$$

Here, the first and second terms represent reconstruction losses for inputs $S_i$ and $S_j$. The third term encourages a minimum distance of $b$ between any pair of encodings to yield unique encodings for different classes. The fourth term is an L2-regularizer.

Once the network is trained, the real-valued encodings $C_i$ are converted to binary encodings such that it has equal numbers of 0s and 1s. This formulation introduces three hyperparameters. We determine the number of bit transitions in the designed label encodings and select hyperparameters that result in the lowest number of bit transitions.

Note that this autoencoder network is decoupled from the regression network and design codes agnostic to classifiers' characteristics for a given regression problem.

## A.3 EXPERIMENTAL METHODOLOGY

We use 11 benchmarks covering four different regression tasks for evaluation. This section summarizes the experimental setup, including datasets, evaluation metrics, hyperparameters, and related work for each of these tasks. We also report the training time using an NVIDIA RTX 2080 Ti GPU with 11GB of memory for each benchmark.

### A.3.1 HEAD POSE ESTIMATION

In landmark-free 2D head pose estimation, for a given 2D image, the head pose of a human is directly estimated in terms of three angles: yaw, pitch, and roll. We use loose cropping around the center with random flipping for data augmentation. We use the ResNet50 network as the feature extractor. This network is initialized using pre-trained parameters for ImageNet (Russakovsky et al., 2015) dataset. During the training for RLEL the entire network, including the feature extractor, is trained.

**Datasets:** We use the evaluation methodology followed by prior works (Ruiz et al., 2018; Yang et al., 2019). Two protocols are used for evaluation.

Protocol 1 (LFH1): This protocol uses the BIWI (Fanelli et al., 2013) dataset for training and evaluation. This dataset consists of $15,128$ frames of 20 subjects. Random $70\% - 30\%$ splits are used for training and evaluation. The ranges of yaw, pitch, and roll angles are $[-75°, 75°]$, $[-65°, 85°]$, and $[-55°, 45°]$, respectively.

Protocol 2 (LFH2): In this protocol, the network is trained using the 300W-LP (Zhu et al., 2016) dataset consisting of $122,450$ samples. AFLW2000 (Zhu et al., 2016) dataset is used for evaluation. The range of all labels is $[-99°, 99°]$ in this setting.

**Evaluation metrics:** We report the Mean Absolute Error (MAE) between the targets $(y_i)$ and predictions $(\hat{y}_i)$. Let $N$ represent the number of samples, and $P$ represent the number of labels (three in head pose estimation). The MAE is defined as:

$$\text{MAE} = \frac{1}{N}\sum_{i=1}^{N}\frac{1}{P}\sum_{j=1}^{P}|y_{i,j} - \hat{y}_{i,j}| \tag{12}$$

**Network architecture and training parameters:** Table 12 summarizes the hyperparameters used for RLEL . The learning rate of the decoding matrix $D$ is kept $10\times$ higher than the learning rate of the feature extractor. L2 regularization with weight of 0.0001 is used for direct regression.

**Related work** Head pose estimation is a widely studied problem. Existing task-specialized approaches propose different loss formulations or feature extractors to improve the error. HopeNet (Ruiz et al., 2018) proposed a combination of regression and classification loss. SSR-Net (Yang et al., 2018) and FSA-Net (Yang et al., 2019) proposed stage-wise soft regression. QuatNet (Hsu et al.,

Table 12: Training parameters for LFH1.

| Approach | Label range/ Quantization levels | Optimizer | Epochs | Batch size | Learning rate | Learning rate schedule | $\beta$ | $\alpha$ | Training time (GPU hours) |
|---|---|---|---|---|---|---|---|---|---|
| LFH1 | Yaw: $[-75°, 75°]/150$, Pitch:$[-65°, 85°]/150$, Roll: $[-55°, 45°]/100$ | Adam, weight decay=0, momentum = 0 | 50 | 8 | 0.0001 | 1/10 after 30 Epochs | 0.5 | 1.0 | 2 |
| LFH2 | $[-99°, 99°]/200$ | Adam, weight decay=0, momentum = 0 | 20 | 16 | 0.00001 | 1/10 after 10 Epochs | 2.0 | 0.0 | 4 |

2019) proposed to use MSE loss with custom ordinal regression loss. RAFA-Net (Behera et al., 2021) proposed an attention-based feature extractor architecture. Table 13 and Table 14 compare the performance of RLEL with related work.

Table 13: Landmark-free 2D Head poses estimation evaluation for protocol 1 (HPE1 and HPE3).

| Approach | Feature Extractor | #Params (M) | Yaw | Pitch | Roll | MAE |
|---|---|---|---|---|---|---|
| SSR-Net-MD (Yang et al., 2018) (Soft regression) | SSR-Net | 1.1 | 4.24 | 4.35 | 4.19 | 4.26 |
| FSA-Caps-Fusion (Yang et al., 2019) (Soft regression) | FSA-Net | 5.1 | _2.89_ | 4.29 | 3.60 | 3.60 |
| RAFA-Net (Behera et al., 2021) (Direct Regression) | RAFA-Net | 69.8 | _3.07_ | 4.30 | _2.82_ | _3.40_ |
| Direct regression (L2 loss) | ResNet50 | 23.5 | 3.80 | 4.63 | 4.28q | $4.22 \pm 0.35$ |
| BEL (Shah et al., 2022) | ResNet50 | 23.6 | **3.32** | 3.80 | **3.53** | $3.56 \pm 0.11$ |
| RLEL | ResNet50 | 23.6 | 3.41 | **_3.20_** | 3.97 | **$3.55 \pm 0.10$** |

Table 14: Landmark-free 2D Head poses estimation evaluation for protocol 2 (HPE2 and HPE4).

| Approach | Feature Extractor | #Params (M) | Yaw | Pitch | Roll | MAE |
|---|---|---|---|---|---|---|
| SSR-Net-MD Yang et al. (2018) (Soft regression) | SSR-Net | 1.1 | 5.14 | 7.09 | 5.89 | 6.01 |
| FSA-Caps-Fusion Yang et al. (2019) (Soft regression) | FSA-Net | 5.1 | 4.50 | 6.08 | 4.64 | 5.07 |
| RAFA-Net Behera et al. (2021) (Direct Regression) | RAFA-Net (HPE4) | 69.8 | 3.60 | 4.92 | 3.88 | _4.13_ |
| HopeNet* ($\alpha = 2$) Ruiz et al. (2018) (classification + regression loss) | ResNet50 | 23.9 | 6.47 | 6.56 | 5.44 | 6.16 |
| Direct regression (L2 loss) | ResNet50 | 23.5 | 5.61 | 6.13 | 4.18 | $5.32 \pm 0.12$ |
| BEL Shah et al. (2022) | ResNet50 | 23.6 | **4.54** | **5.76** | 3.96 | **$4.77 \pm 0.05$** |
| RLEL | ResNet50 | 23.6 | 4.69 | 5.79 | **3.86** | **$4.77 \pm 0.05$** |

### A.3.2 FACIAL LANDMARK DETECTION

Facial landmark detection focuses on finding the $(x, y)$ coordinates of facial keypoints for a given 2D image.

**Evaluation metrics:** We report the Normalized Mean Error (NME) between the targets $y_i$ and predictions $\hat{y}_i$. Inter-ocular distance normalization is used for all datasets. For $N$ test samples, $P$ facial landmarks, and $L$ normalization factor, the NME is defined as:

$$\text{NME} = \frac{1}{N} \sum_{i=1}^{N} \frac{1}{P} \cdot \frac{1}{L} \sum_{j=1}^{P} |y_{i,j} - \hat{y}_{i,j}|_2 \qquad (13)$$

**Datasets:** We use three datasets widely used for facial landmark detection: COFW (Burgos-Artizzu et al., 2013), 300W (Sagonas et al., 2013), and WFLW (Wu et al., 2018). HRNetV2-W18 network architecture for feature extraction (Wang et al., 2020) and the modified regressor architecture for

label encoding proposed by BEL (Shah et al., 2022) are used in this work. Random flipping, scaling $(0.75 - 1.25)$, and rotation $(\pm 30)$ are used for data augmentation. The COFW dataset consists of $1,345$ training and $507$ testing images annotated with 29 landmarks. The training set of the 300W dataset has $3,148$ images annotated with 68 facial landmarks. This dataset provides four test sets: full test set, common subset, challenging subset, and the official test set with 300 indoor and 300 outdoor images. WFLW dataset is a comparatively larger dataset with $7,500$ training and $2,500$ testing images. Each image is annotated with 98 facial landmarks. The test set is divided into six subsets: large pose, expression, illumination, make-up, occlusion, and blur.

**Training parameters:** Table 15 provides a summary of all the training parameters. The learning rate of the decoding matrix $D$ is kept $20\times$ higher than the learning rate of the feature extractor. The HRNetV2-W18 network is initialized with pretrained weight parameters for the ImageNet dataset. We refer to HRNetV2-W18 evaluated on COFW as **FLD1/FLD1_s**, on 300W as **FLD2/FLD2_s**, and on WFLW as **FLD3/FLD3_s**.

Table 15: Training parameters for facial landmark detection for HRNetV2-W18 feature extractor.

| Dataset/ Benchmark | Optimizer | Epochs | Batch size | Learning rate (BEL/Direct regression/Multiclass classification) | Learning rate schedule | $\beta$ | $\alpha$ | Training time (GPU hours) |
|---|---|---|---|---|---|---|---|---|
| COFW/ FLD1 | Adam, weight decay=0, momentum = 0 | 60 | 8 | 0.0005/0.0003/ 0.0003 | 1/10 after 30 and 50 Epochs | 3.0 | 0.0 | $\frac{1}{2}$ |
| COFW/ FLD1_s | Adam, weight decay=0, momentum = 0 | 60 | 8 | 0.0005/0.0003/ 0.0003 | 1/10 after 30 and 50 Epochs | 4.0 | 0.0 | $\frac{1}{2}$ |
| 300W/ FLD2 | Adam, weight decay=0, momentum = 0 | 60 | 8 | 0.0007/0.0003/ 0.0003 | 1/10 after 30 and 50 Epochs | 5.0 | 1.0 | 3 |
| 300W/ FLD2_s | Adam, weight decay=0, momentum = 0 | 60 | 8 | 0.0007/0.0003/ 0.0003 | 1/10 after 30 and 50 Epochs | 5.0 | 0.05 | 3 |
| WFLW/ FLD3 | Adam, weight decay=0, momentum = 0 | 60 | 8 | 0.0003/0.0003/ 0.0003 | 1/10 after 30 and 50 Epochs | 0.0 | 0.1 | 5 |
| WFLW/ FLD3_s | Adam, weight decay=0, momentum = 0 | 60 | 8 | 0.0003/0.0003/ 0.0003 | 1/10 after 30 and 50 Epochs | 5.0 | 0.1 | 5 |

**Related work** Facial landmark detection is a widely studied problem. A common approach is to use heatmap regression, where the target heatmaps are generated by assuming a Gaussian distribution around the ground truth landmark location. Prior works proposed the use of binary heatmaps with pixel-wise binary cross-entropy loss (Bulat & Tzimiropoulos, 2016). HRNet (Wang et al., 2020) proposed a feature extractor that maintains high-resolution representations and uses heatmap regression. AWing (Wang et al., 2019) proposed a modified heatmap regression loss function with adapted wing loss. AnchorFace (Xu et al., 2020) used anchoring of facial landmarks on templates. LUVLi (Kumar et al., 2020) proposed a landmark's location, uncertainty, and visibility likelihood-based loss. Table 16- 18 compare RLEL with related work.

Table 16: Facial landmark detection results on COFW dataset (FLD1).

| Approach | Feature Extractor | #Params/ GFlops | Test NME | $FR_{0.1}$ |
|---|---|---|---|---|
| LAB (w B) (Wu et al., 2018) | Hourglass | 25.1/19.1 | 3.92 | 0.39 |
| AWing (Wang et al., 2019)* | Hourglass | 25.1/19.1 | 4.94 | - |
| HRNetV2-W18 (Wang et al., 2020) (Heatmap regression) | HRNetV2-W18 | 9.6/4.6 | 3.45 | 0.19 |
| Direct regression (L2 loss) | HRNetV2-W18 | 10.2/4.7 | $3.96 \pm 0.02$ | 0.29 |
| Direct regression (L1 loss) | HRNetV2-W18 | 10.2/4.7 | $3.60 \pm 0.02$ | 0.29 |
| BEL (Shah et al., 2022) | HRNetV2-W18 | 10.6/4.6 | $\mathbf{3.34} \pm 0.02$ | 0.40 |
| RLEL | HRNetV2-W18 | 10.6/4.6 | $3.36 \pm 0.01$ | 0.20 |

∗Uses different data augmentation for the training

Table 17: Facial landmark detection results on 300W dataset (FLD2).

| Approach | Feature Extractor | #Params/ GFlops | Test | Common | Challenging | Full |
|---|---|---|---|---|---|---|
| DAN (Kowalski et al., 2017) | - | - | - | 3.19 | 5.24 | 3.59 |
| LAB (w B) (Wu et al., 2018) | Hourglass | 25.1/19.1 | - | 2.98 | 5.19 | 3.49 |
| AnchorFace (Xu et al., 2020) | ShuffleNet-V2 | - | - | 3.12 | 6.19 | 3.72 |
| AWing (Wang et al., 2019)* | Hourglass | 25.1/19.1 | - | 2.72 | 4.52 | 3.07 |
| LUVLi (Kumar et al., 2020) | CU-Net | - | - | 2.76 | 5.16 | 3.23 |
| HRNetV2-W18 (Wang et al., 2020) (Heatmap regression) | HRNetV2-W18 | 9.6/4.6 | - | **2.87** | **5.15** | **3.32** |
| Direct regression (L2 loss) | HRNetV2-W18 | 10.2/4.7 | 4.40 | 3.25 | 5.65 | 3.71 ± 0.05 |
| Direct regression (L1 loss) | HRNetV2-W18 | 10.2/4.7 | 4.26 | 3.10 | 5.42 | 3.54 ± 0.03 |
| BEL (Shah et al., 2022) | HRNetV2-W18 | 11.2/4.6 | 4.09 | 2.91 | 5.50 | 3.40 ± 0.02 |
| RLEL | HRNetV2-W18 | 11.2/4.6 | **4.03** | 2.90 | 5.39 | 3.37 ± 0.02 |

*Uses different data augmentation for the training

Table 18: Facial landmark detection results (NME) on WFLW test (FLD3) and 6 subsets: pose, expression (expr.), illumination (illu.), make-up (mu.), occlusion (occu.) and blur. $\theta = 10$ is used for BEL.

| Approach | Feature Extractor | #Params/ GFlops | Test | Pose | Expr. | Illu. | MU | Occu. | Blur |
|---|---|---|---|---|---|---|---|---|---|
| LAB (w B) (Wu et al., 2018) | Hourglass | 25.1/19.1 | 5.27 | 10.24 | 5.51 | 5.23 | 5.15 | 6.79 | 6.32 |
| AnchorFace (Xu et al., 2020)* | HRNetV2-W18 | -/5.3 | 4.32 | 7.51 | 4.69 | 4.20 | 4.11 | 4.98 | 4.82 |
| AWing (Wang et al., 2019)* | Hourglass | 25.1/19.1 | 4.36 | 7.38 | 4.58 | 4.32 | 4.27 | 5.19 | 4.96 |
| LUVLi (Kumar et al., 2020) | CU-Net | - | 4.37 | - | - | - | - | - | - |
| HRNetV2-W18 (Wang et al., 2020) (Heatmap regression) | HRNetV2-W18 | 9.6/4.6 | 4.60 | 7.94 | 4.85 | 4.55 | 4.29 | 5.44 | 5.42 |
| Direct regression (L1 loss) | HRNetV2-W18 | 10.2/4.7 | 4.64 ± 0.03 | 8.13 | 4.96 | 4.49 | 4.45 | 5.41 | 5.25 |
| BEL (Shah et al., 2022) | HRNetV2-W18 | 11.7/4.6 | 4.36 ± 0.02 | **7.53** | 4.64 | **4.28** | 4.19 | 5.19 | **5.05** |
| RLEL | HRNetV2-W18 | 11.7/4.6 | **4.35 ± 0.01** | 7.57 | **4.57** | 4.36 | 4.19 | 5.25 | 5.07 |

*Uses different data augmentation for the training

### A.3.3 AGE ESTIMATION

This task focuses on predicting a person's age from a given 2D image. MAE (Equation 12) and Cumulative Score (CS) are used as the evaluation metrics, and ResNet50 (He et al., 2016) is used as the feature extractor. CS$\theta$ is the percentage of test samples with absolute error less than $\theta$ years.

**Datasets**  MORPH-II (Ricanek & Tesafaye, 2006) and AFAD (Niu et al., 2016) datasets are used for evaluation. We follow the protocols for preprocessing and data augmentation of datasets as per prior works (Shah et al., 2022; Raschka, 2018). MORPH-II dataset consists of $55,608$ images with random split of $39,617$ training, $4,398$ validation, and $11,001$ test images. The AFAD dataset consists of $164,432$ images with random split of $118,492$ training, $13,166$ validation, and $32,763$ test images.

**Training parameters:**  Table 19 summarizes the training parameters for **AE1** (MORPH-II) and **AE2** (AFAD) benchmarks. The learning rate of the decoding matrix $D$ is kept $10\times$ higher than the learning rate of the feature extractor. L2 regularization with weight of 0.001 is used for direct regression. Training for AE1 and AE2 consumes $\sim 2$ and $\sim 7$ hours, respectively.

Table 19: Training parameters for age estimation using MORPH-II and AFAD dataset

| Bench-mark | Optimizer | Epochs | Batch size | Learning rate | Learning rate schedule | $\beta$ | $\alpha$ |
|---|---|---|---|---|---|---|---|
| AE1 | Adam, weight decay=0, momentum=0 | 50 | 64 | 0.0001 | - | 0.0 | 2.0 |
| AE2 | Adam, weight decay=0, momentum=0 | 50 | 64 | 0.0001 | - | 0.0 | 5.0 |

**Related work** Different approaches including ordinal regression (Niu et al., 2016; Cao et al., 2020; Pan et al., 2018; Gao et al., 2018), soft stage-wise regression (Yang et al., 2018; 2019), soft labels (Díaz & Marathe, 2019) have been proposed for age estimation. OR-CNN (Niu et al., 2016) and CORAL-CNN (Cao et al., 2020) proposed ordinal regression by binary classification with threshold-based encodings (i.e., unary codes). DLDL (Gao et al., 2018) augmented the loss function with KL-divergence between softmax output and soft target probability distributions. MV-Loss (Pan et al., 2018) proposed to penalize the prediction based on the variance of the age distribution. We compare CLL with related work in Table 20 and Table 21.

Table 20: Age estimation results on MORPH-II dataset (AE1).

| Approach | Feature extractor | #Parameters (M) | MORPH-II (MAE) | MORPH-II ($CS\theta = 5$) |
|---|---|---|---|---|
| OR-CNN (Niu et al., 2016) (Ordinal regression by binary classification ) | - | 1.0 | 2.58 | 0.71 |
| MV Loss (Pan et al., 2018) (Direct regression) | VGG-16 | 138.4 | 2.41 | 0.889 |
| DLDL-v2 (Gao et al., 2018) (Ordinal regression with multi-class classification) | ThinAgeNet | 3.7 | 1.96* | - |
| CORAL-CNN (Cao et al., 2020) (Ordinal regression by binary classification) | ResNet34 | 21.3 | 2.49 | - |
| Direct Regression (L2 Loss) | ResNet50 | 23.1 | $2.37 \pm 0.01$ | $0.903 \pm 0.002$ |
| BEL (Shah et al., 2022) | ResNet50 | 23.1 | $\mathbf{2.27} \pm 0.01$ | $\underline{\mathbf{0.928}} \pm 0.001$ |
| RLEL | ResNet50 | 23.1 | $2.28 \pm 0.01$ | $0.901 \pm 0.002$ |

Table 21: Age estimation results on AFAD dataset (AE2).

| Approach | Feature extractor | #Parameters (M) | AFAD (MAE) | AFAD ($CS\theta = 5$) |
|---|---|---|---|---|
| OR-CNN (Niu et al., 2016) (Ordinal regression by binary classification ) | - | 1.0 | 3.51 | 0.74 |
| CORAL-CNN (Cao et al., 2020) (Ordinal regression by binary classification) | ResNet34 | 21.3 | 3.47 | - |
| Direct Regression (L2 Loss) | ResNet50 | 23.1 | $3.16 \pm 0.02$ | $0.810 \pm 0.02$ |
| BEL (Shah et al., 2022) | ResNet50 | 23.1 | $\mathbf{3.11} \pm 0.01$ | $\underline{\mathbf{0.823}} \pm 0.001$ |
| RLEL | ResNet50 | 23.1 | $3.14 \pm 0.01$ | $80.78 \pm 0.002$ |

### A.3.4 END-TO-END SELF DRIVING

For the regression task of end-to-end autonomous driving, we use the NVIDIA PilotNet dataset, and PilotNet model (Bojarski et al., 2016). In this task, for a given image of the road, the angle of the steering wheel that should be taken next is predicted. MAE (Equation 12) is used as the evaluation metric.

**Dataset** The PilotNet driving dataset consists of $45,500$ images taken around Rancho Palos Verdes and San Pedro, California (Chen). We use the data augmentation technique used by prior works (Bojarski et al., 2016).

**Training parameters** Table 22 summarizes the training parameters. The learning rate of the decoding matrix $D$ is kept $10\times$ higher than the learning rate of the feature extractor.

Table 22: Training parameters for end-to-end autonomous driving using PilotNet.

| Optimizer | Epochs | Batch size | Learning rate | Learning rate schedule | $\beta$ | $\alpha$ |
|---|---|---|---|---|---|---|
| SGD with weight decay=1e-5, momentum=0 | 50 | 64 | 0.1 | 1/10 at 10, 30 epochs | 0.0 | 2.0 |

**Related work** End-to-end autonomous driving is an interesting task with increasing attention. PilotNet (Bojarski et al., 2017) used a small, application-specific network. We compare RLEL with the baseline PilotNet architecture in Table 23.

Table 23: End-to-end autonomous driving results on PilotNet dataset (PN) and architecture (Bojarski et al., 2017; 2016).

| Approach | Feature extractor | #Parameters (M) | MAE |
|---|---|---|---|
| PilotNet (Bojarski et al., 2017) | PilotNet | 1.8 | $4.24 \pm 0.45$ |
| BEL (Shah et al., 2022) | PilotNet | 1.8 | $3.11 \pm 0.01$ |
| RLEL | PilotNet | 1.8 | $\mathbf{2.94} \pm 0.01$ |

