# OpenReview forum: "Learning Label Encodings for Deep Regression"
_ICLR.cc/2023/Conference — ICLR 2023 notable top 25%_

### Official Review · Reviewer_XTEa · 2022-10-22

**Confidence:** 2
**Correctness:** 4
**Technical Novelty And Significance:** 3
**Empirical Novelty And Significance:** 3
**Recommendation:** 6

**Clarity, Quality, Novelty And Reproducibility:**

Clarity:

The manuscript is well-presented and the idea is simple and easy to follow.

Quality:

The method is extensively demonstrated on 11 benchmarks with significant improvement upon baselines. The ablation study also demonstrates the effectiveness of the proposed regularizers.

Reproducibility:

The authors have their code attached, and I think the implementation details provided in the appendix are sufficient.

**Strength And Weaknesses:**

strength:

(1) The authors tackle the problem of deep regression through the lens of binary classification. It is also interesting and technically sound to extend the conventional hand-crafted encoding method to be differentiable. Overall, I like this simple yet useful idea.

(2) The experiments are extensive, and the proposed end-to-end encoding method has competitive performance compared to hand-crafted. The ablation study also demonstrates the effectiveness of two proposed regularizers.

Weakness:

(1) In the experiments, manually designed BEL outperforms RLEL in many cases. I'm curious why an end-to-end training method seemingly achieves a suboptimal solution, compared to a hand-crafted method. Can the authors provide some intuition on it? Also, how different is it between the learned encodings and (better performed) hand-crafted encodings?

(2) I encourage the authors to discuss the efficiency of the method. What is the time cost to incorporate the label encoding learning with regularizers, compared to other hand-crafted encodings?

**Summary Of The Paper:**

This paper introduces Regularized Label Encoding Learning (RLEL) for end-to-end training of an entire network and its label encoding, by combining continuous label encodings space with regularizers. The proposed method is extensively demonstrated on 11 benchmarks.

**Summary Of The Review:**

The authors propose a simple yet effective end-to-end training to simultaneously learn the label encoding and features. The experiments are solid and extensively demonstrate the effectiveness of the method. Therefore, I recommend borderline acceptance.

---

> ### Author Response · Authors · 2022-11-19
> **Author response**
>
>
> Thank you for your thoughtful comments and feedback. We have updated the paper and supplemental material to reflect the response below.
>
>
> **Q1. Comparison of RLEL and BEL**
>
> BEL [1] explores different training loss functions as well. It explores cross-entropy loss, binary cross-entropy loss, and L1/L2 loss for all benchmarks, and the loss function resulting in the lowest validation error is reported in Table 2. In contrast, RLEL focuses on learning label encodings and uses cross-entropy loss (Section 3.3) for all benchmarks. Thus, in the result reported in Table 2, BEL explores more loss functions than RLEL.
>
> We also study the effect of dataset size on BEL and RLEL approaches. We train BEL and RLEL with the same training loss function (cross entropy loss in Eq. (5)). We take the dataset FLD1 and use a fraction of the dataset for training. The entire test dataset is used for testing here. The table below summarizes the error achieved by RLEL and BEL for different fractions of the training dataset. The evaluation shows that the gap between the performance of RLEL and BEL decreases with the increase in dataset size, which suggests that RLEL might be able to achieve lower error for larger datasets.
>
> | %Dataset used | RLEL | BEL  | Difference (RLEL-BEL) |
> |---------------|------|------|-----------------------|
> |           100 | 3.36 | 3.35 |                  0.01 |
> |            80 | 3.43 | 3.42 |                  0.01 |
> |            60 | 3.53 | 3.47 |                  0.06 |
> |            40 | 3.77 | 3.72 |                  0.05 |
> |            20 | 4.08 | 4.04 |                  0.04 |
> |            10 | 4.71 | 4.63 |                  0.08 |
>
> **Changes in the paper:** We have clarified in Section 4 that BEL explores more loss functions. We have also added the above results in Section A.1.9.
>
> <hr>
>
> **Q2. Comparison of learned and hand-crafted encodings**
>
> We visually compare the encoding learned by RLEL with BEL hand-crafted code for one benchmark (Figure 10 in Section A.1.10 in the revised paper). We notice some common characteristics between both encodings. For example, the codes for nearby labels differ by fewer bits than faraway labels. Both the codes also have fewer bit transitions ($0\rightarrow1$ and $1\rightarrow0$ transitions in a column). These characteristics in the learned encodings are encouraged by the proposed regularizers R1 and R2. There are a few differences between learned and hand-crafted encodings. In contrast to hand-crafted labels, encodings for adjacent labels do not differ in some cases. Hand-crafted encoding assures at least one or two bits of difference between adjacent labels.
>
> **Changes in the paper:** We have added Section A.1.10 to compare learned and hand-crafted encodings.
>
> **Q3. Time-cost to incorporate label encoding learning compared to hand-crafted codes**
>
> Training time:
> In label encoding learning, the feature extractor and label encodings are trained together end-to-end. The networks are trained for the same number of epochs for label encoding learning with regularizers and hand-crafted encodings. Similarly, the proposed regularizer does not add significant overhead to the computation (less than 0.01%). Hence, the training time for regularized label encoding learning and hand-crafted encodings is the same.
>
> Regularized label encoding learning requires hyperparameter tuning for the regularizer parameters. Several approaches have been proposed for efficient hyperparameter search [2][3], and any off-the-shelf hypermeter tuners/libraries can be used to automatically find these values without manual efforts. In contrast, hand-designed codes need human intervention to design codes. Also, multiple training runs are still required to find suitable codes for a given benchmark from a set of hand-designed codes. On the other hand, RLEL provides an end-to-end automated approach for label encoding learning.
>
> **Changes in the paper:** We have clarified in Section 4.1 that the training time for label-encoding learning is the same as using hand-crafted codes.
>
> <hr>
>
> **References**
> [1] Deval Shah, Zi Yu Xue, and Tor M. Aamodt. Label encoding for regression networks. In International Conference on Learning Representations, April 2022. URL https://openreview.net/pdf?id=8WawVDdKqlL.
>
> [2] Lisha Li, Kevin Jamieson, Giulia DeSalvo, Afshin Rostamizadeh, and Ameet Talwalkar. Hyperband: A novel bandit-based approach to hyperparameter optimization. J. Mach. Learn. Res., 18(1): 6765–6816, jan 2017. ISSN 1532-4435.
>
> [3] Stefan Falkner, Aaron Klein, and Frank Hutter. Bohb: Robust and efficient hyperparameter optimization at scale. In ICML, 2018.

---

### Official Review · Reviewer_TMxF · 2022-10-23

**Confidence:** 3
**Correctness:** 3
**Technical Novelty And Significance:** 2
**Empirical Novelty And Significance:** 3
**Recommendation:** 8

**Clarity, Quality, Novelty And Reproducibility:**

This paper is written well and organized well. Although some details and discussions of the number of soft target labels N are missing, this paper has done well in other aspects.

**Strength And Weaknesses:**

Strengths:
- This paper proposes a novel label encoding method to solve regression and seems can alleviate the disadvantage of the previous label encoding method, the space of label encodings for regression is large.
- This paper introduces two simple yet meaningful rules for label encoding and designs corresponding regularization terms to encourage label encodings to satisfy the rules.
- Comprehensive experimental results on various datasets clearly demonstrate the effectiveness of the proposed method.

Weaknesses:
- According to Eq. (5), the generation of soft target labels is affected by the batch size K. It seems impossible. So, I guess it should be the number of target labels N rather than K in the generation of soft target labels.
- The authors should use unified representation. For example, the correlation vector is denoted by $\hat{C}_i$ in most cases. However, $C_i$ is used in Eq. (5) and Fig. 3.
- The authors assume $y_i \in [1, N]$ for simplicity as the real-valued targets with any arbitrary numeric range can be scaled and shifted to this range. However, they do not provide details and discussions of N. Obviously, different N would cause different performance.



**Summary Of The Paper:**

This paper focuses on deep regression by using label encoding. Specifically, this paper proposes Regularized Label Encoding Learning (RLEL) for end-to-end training of an entire network and its label encodings. RLEL contains two regularization terms, designed to encourage encodings with certain properties. The first one encourages the rule that the L1 distance between label encodings should increase as the difference between two labels increases. The second one encourages the rule that the L1 distance between label encodings for adjacent target label values should be minimized. The effectiveness of the proposed method is demonstrated by a lot of experiments.

**Summary Of The Review:**

Overall, this paper proposes a novel label encoding method that is reasonable to solve the deep regression problem. There are also some shortcomings in this paper as mentioned above. The authors should carefully check the manuscript repeatedly and provide necessary details and discussions to further polish this paper.

---

> ### Author Response · Authors · 2022-11-19
> **Author response**
>
>
> Thank you for your thoughtful comments and feedback. We have updated the paper and supplemental material to reflect the response below.
>
> **Q1. Correction on soft target labels generation in Eq. (5)**
>
> Thank you for pointing out this. There is a typo in that equation. The number of target labels N is used for soft target label generation. We have modified Eq. (5) to correct this.
>
> **Changes in the paper:** Corrected Eq. (5).
>
>
>  **Q2. Notation for correlation vector and unified notation**
>
> We have modified Eq. (5) and Figure 3 to use $\hat{C}_i$ instead of $C_i$. We have also added Table 1 to summarize useful notation in Section 3 for clarity.
>
> **Changes in the paper**: Corrected Eq. (5) and Figure 3. We have added Table 1 to summarize all the notations (Table 1 in Section 3.1 on Page 5).
>
> **Discussion on the value of N**
>
> The number of quantization buckets is treated as a design parameter for binary-encoded labels. Shah et al. [1] showed that the error changes with the number of quantization levels. Fewer levels introduce quantization error, and more levels increase the number of bits in the encoding. They showed a trade-off between these two factors to decide the number of quantization levels.
>
>  Our work focuses on the design space of encoding and decoding functions. Hence we use the same values for the quantization levels ($N$) as BEL [1]. Parameter $N$ tuning can be integrated into hyperparameter tuning or included in the optimization process.
>
> We further analyze the effect of the number of quantization levels for RLEL. The following table shows the NME (Normalized Mean Error) for different values of $N$ for FLD1 benchmark.
> | Quantization levels (N) | NME  |
> |-------------------------|------|
> |                      32 | 3.49 |
> |                      64 | 3.36 |
> |                     128 | 3.36 |
> |                     256 | 3.36 |
> |                     384 | 3.37 |
> |                     512 | 3.37 |
>
> This suggests that the proposed method RLEL is less sensitive to the number of quantization levels for higher values. For RLEL, the decoding matrix that converts the label encodings to the predicted label is also learned during the training (Figure 3). This matrix is of size $M\times N$, where each column represents the weight parameters for one quantization level. One possible reason for the above results is that matrix $D$ adaptively learns the number of quantization levels suitable for this problem.
>
>
> **Changes in the paper:** Added results for different quantization levels in Section A.1.8. We have also clarified how we select the value of $N$ in Section 4.1.
>
> <hr>
>
> **References**
>
> [1] Deval Shah, Zi Yu Xue, and Tor M. Aamodt. Label encoding for regression networks. In International Conference on Learning Representations, April 2022. URL https://openreview.net/pdf?id=8WawVDdKqlL.

---

> > ### Comment · Reviewer_TMxF · 2022-11-22
> > **Thanks**
> >
> > Thanks for the rebuttal. I am satisfied with it and I checked the comments from other reviewers and decide to increase my score.

---

### Official Review · Reviewer_ZxJ5 · 2022-10-23

**Confidence:** 4
**Correctness:** 3
**Technical Novelty And Significance:** 3
**Empirical Novelty And Significance:** 2
**Recommendation:** 8

**Clarity, Quality, Novelty And Reproducibility:**

The quality of writing is excellent. The author provides extensive experiment setting and code in the appendix and supplemental materials. I believe it will be sufficient to reproduce the work.

Minor comment:
In figure 1(a), the legend (the blue dot vs red dot is too small) is almost unrecognizable.


**Strength And Weaknesses:**

Strength: clarity and quality of writing. Extensive experiments.

The paper is well written with a pretty clear introduction of background and related work. I also like how the author motivates the underlying work by showing the challenges in the design space. By giving a literature review on the binary encoding's usage in regressor, I was able to understand why the properties of suitable encodings (identified by prior works) are important in such a work. By following this guidance to the properties, the author started to introduce the regularization functions to encourage such properties when using real-valued label encodings. The loss function is then introduced based on the regularizer design. The experiments use almost the same datasets (from Shah 2022) so that a fair comparison with its predecessor can be easily conducted. The author also compares with a few other label encoding design approaches and regression approach to show that their work is a generic regression approach and generalizes better with even a small dataset.

Weakness: not strong originality.

Most of the originality in this paper is to propose a learned continuous label encoding (compared to existing work on using manually designed binary encoding). From the regularizer design, the author identifies effective regularization functions to encourage the expected property within a continuous encoder. The proposed method seems effective but incremental in nature.

Questions to the author:
1. How would you decide the number of quantization buckets ($M$ in the continuous predicted encoding $Z_i$)? Is there an optimal dimension size for various dataset?
2. Computationally (during training or inference), is there any significant difference between RLEL and BEL?
3. In equation (3), can you give more explanation that why you choose the magic number 2?
4. It seems to me (by looking at Table2) BEL is reasonably performing well though it is manually designed. I kindly expect that adaptively learned encoder shall perform even better if more data is given during training. I am not familiar with the scale of the experimental dataset you have used. How many examples are there in each of them? Were you able to test your method in a significantly large-scaled problem (e.g., with many more training examples) to show the superiority of your model)?
5. Does RLEL use uniform quantization (similar to BEL)? Is there something that can be done to adaptively select quantization level and non-uniform quantization?

**Summary Of The Paper:**

This paper proposes an end-to-end automated approach to find label encodings for deep regression problem. In my understanding, these tasks are usually using images as input and are outputting a continuous value. Due to the nature of deep regression, a general approach to regress directly does not show good results compared to task-specific approaches. Their work is largely based on a prior work [Shah 2022], in which 1) binary-encoded labels were introduced to encode quantized regression values 2) three desirable properties of suitable encoding/decoding functions were identified. This paper extends [Shah 2022]'s work by not using manually identified binary codes, but learn the continuous label encodings from the data directly. Their empirical experiments show that their learned-encodings and the new model result in lower or comparable errors to [Shah 2022] work.

[Shah 2022] Deval Shah, Zi Yu Xue & Tor M. Aamodt. LABEL ENCODING For REGRESSION NETWORKS. ICLR 2022.



**Summary Of The Review:**

Weak accept. The paper proposes an effective way to automatically learn continuous encoding to help improve deep regression model performance. If the author could give more theoretical analysis on why the proposed model encourages the expected encoder properties, I would be happy to increase my scores.

---

> ### Author Response · Authors · 2022-11-19
> **Author response (Part 3)**
>
> **Q6. Theoretical analysis of why the proposed model encourages the expected encoder properties**
>
> **Regularizer R2:**
>
> We proposed to use the regularization function Eq. (4) to encourage the properties for label encodings shown in Eq. (2). We proposed to use matrix $D$ to impose the expected properties of encodings $E$.
>
> We insight into this decision as follows. First, note the output label encodings are multiplied with $D$ to generate the correlation vector $\hat{C}_i$ (Figure 3). We use the multiclass classification loss between $\hat{C}_i$ and the target labels for training. Due to this, label encodings $E$ and decoding matrix $D$ are related, and use of matrix $D$ proves to be effective for regularizer R2. We further explain this in detail below.
>
> Let $E$ represent an encoding matrix of size $N\times M$. Each row $E_{k:}$ represents the label encoding output when the label is $k$. $D$ is the decoding matrix of size $M\times N$. Let $\hat{C}\_{k}$ represent the output correlation row vector of size $1\times N$ when the target label is $k$. Here, $\hat{C}\_{k}$ is obtained by multiplying $E_{k,:}$ with $D$ (Figure 3).
>
> $\hat{C}_{k} = E_{k,:} D$    (6)
>
> Since we apply softmax on the output vector to find the predicted label (Figure 3), ideally, $\hat{C}^k_{k}$ should have the highest value as the target label value is $k$.
>
> $\therefore \hat{C}^k_{k} > \hat{C}^x_{k} ,\text{ where, } x \neq k, x\in  \{1,2,...,N\}$.
>
> $\therefore E_{k,:} . D_{:,k} > E_{k,:} . D_{:,x},\text{ where, } x \neq k, x\in  \{1,2,...,N\}$ (Using Eq. (6))
>
> Let $\theta_{k,x}$ represents the angle between row vector $E_{k,:}$ and column vector $D_{: ,x}$. This leads to the below equation:
>
> $||E_{k,:}|| ||D_{:,k}|| \text{cos}(\theta_{k,k}) > ||E_{k,:}|| ||D_{:,x}|| \text{cos}(\theta_{k,x}),\text{ where, } x \neq k, x\in  \{1,2,...,N\}$ (7)
>
> Shah et al. [1] used a handcrafted decoding matrix $D$ with an equal number of $1$s and $0$s in each column for binary-encoded labels. Hence the L2 norm of each column is the same. In label encoding learning, parameters of matrix $D$ are learned during training and are not constrained to have the same L2 norm for each column. However, we observe a similar trend empirically. Figure 9 (in the paper) plots the distribution of $||D_{:,x}||$ for different benchmarks. As shown in the figure, for most benchmarks, we observe a small variance in the distribution of $||D_{:,x}||$. For example, for LFH2 benchmark, the mean and variance of  $||D_{:,x}||$ are $8.18$ and $0.061$, respectively .Based on this intuition and empirical validation, we assume that $||D_{:,x}|| \approx ||D_{:,y}||$ for $x\in[1, N]$ and $y\in[1, N]$ to simplify the analysis.
>
> Using this assumption in Eq. (7) leads to the following inequality:
>
> $\text{cos}(\theta_{k,k}) > \text{cos}(\theta_{k,x}),\text{ where, } x \neq k, x\in  \{1,2,...,N\}$
>
> Thus the cosine similarity between $E_{k,:}$ and $D_{:,k}$ should be the highest to predict label $k$. The optimization process to reduce the loss between the target and prediction will try to maximize this cosine similarity. In the best case, the angle between $E_{k,:}$ and $D_{:,k}$ will be zero, and both vectors are parallel.
>
> This simplification leads to the following relation between $E$ and $D$.
>
> $E_{k,:} = t D_{:,k}$ ,  where $t > 0$
>
> Similarly, $E_{k+1,:} = t' D_{:,k+1}$ , where $t' > 0$
>
> Since $t$ and $t'$ both are positive values, reducing $D_{i,k}-D_{i,k+1}$ also reduces $E_{k,i}-E_{k+1,i}$.
>
> Regularizer rule R2 proposes to regularize the number of decision boundaries by regularizing $\sum_{i=1}^{M} \sum_{j=1}^{N-1} |{E}_{j,i} - E_{j+1,i}|$ as per Eq. (2).
>
> Based on the analysis above, regularizig $\sum_{i=1}^{M} \sum_{j=1}^{N-1} |D_{i,j}-D_{i,j+1} |$ helps with the above goal as $E_{j,i}-E_{j+1,i}$ reduces with $D_{i,j}-D_{i,j+1}$.
>
>
> **Regularizer R1**
>
> The first property suggests $||E_{i,:} - E_{j,:}||_1 \propto |i-j|$.
>
> So ideally, $||E_{i,:} - E_{j,:}||_1 = \lambda |i-j|$
>
> Since $E_{x,:}$ is an average of $\hat{Z}_i$ for samples with label value $x$ (Eq. (1)), the above condition leads to:
>
> $||\hat{Z}\_{i} - \hat{Z}\_{j}||_1 = \lambda |y_i-y_j|$ (8)
>
> Based on this requirement, we add a regularization function max$(0,\lambda |y_i - y_j | - ||\hat{Z}_i - \hat{Z}_j ||_1)$, which penalizes the label encodings if $||\hat{Z}_i - \hat{Z}_j ||_1$ < $\lambda |y_i-y_j|$. It does not strictly impose the above equality (Eq. (8)). However, it approximately imposes the constraint as per shown in empirical verification in Section A.1.1.
>
> We set $\lambda=2$ as explained in Q3.
>
> **Changes in the paper:** We have added Section A.1.7 to add this theoretical analysis.
>
> <hr>
>
> **References**
>
> [1] Deval Shah, Zi Yu Xue, and Tor M. Aamodt. Label encoding for regression networks. In International Conference on Learning Representations, April 2022. URL https://openreview.net/pdf?id=8WawVDdKqlL.
>
> [2] https://mmclassification.readthedocs.io/en/latest/model_zoo.html

---

> ### Author Response · Authors · 2022-11-19
> **Author response (Part 2)**
>
> **Q4. Comparison of RLEL and BEL; effect of the training dataset size**
>
> Experimental datasets used in this work are of varying scales. We summarize the number of training images for all benchmarks:
>
> LFH1: $10589$, LFH2: $122450$, FLD1: $1345$, FLD1_s: $134$, FLD2: $3148$, FLD2_s: $314$, FLD3: $7500$, FLD3_s: $750$, AE1: $396174, AE2: $164432$, PN: $45500$
>
> BEL [1] explores different training loss functions as well. It explores cross-entropy loss, binary cross-entropy loss, and L1/L2 loss for all benchmarks, and the loss function resulting in the lowest validation error is reported in Table 2. In contrast, RLEL focuses on learning label encodings and uses cross-entropy loss (Section 3.3) for all benchmarks. Thus, in the result reported in Table 2, BEL explores more loss functions than RLEL.
>
> In order to compare the effect of dataset size on encoding design, we run BEL and RLEL approaches with the same training loss function (cross entropy loss in Eq. (5)). We take the dataset FLD1 and use a fraction of the dataset for training. The entire test dataset is used for testing here. The table below summarizes the error achieved by RLEL and BEL for different fractions of the training dataset. The evaluation shows that the gap between the performance of RLEL and BEL decreases with the increase in dataset size, which suggests that RLEL might be able to achieve lower error for larger datasets.
>
> | %Dataset used | RLEL | BEL  | Difference (RLEL-BEL) |
> |---------------|------|------|-----------------------|
> |           100 | 3.36 | 3.35 |                  0.01 |
> |            80 | 3.43 | 3.42 |                  0.01 |
> |            60 | 3.53 | 3.47 |                  0.06 |
> |            40 | 3.77 | 3.72 |                  0.05 |
> |            20 | 4.08 | 4.04 |                  0.04 |
> |            10 | 4.71 | 4.63 |                  0.08 |
>
> **Changes in the paper:** We have clarified in Section 4 that BEL explores more loss functions. We have also added the above results in Section A.1.9.
>
> <hr>
>
> **Q5. Adaptively selecting quantization level and non-uniform quantization**
>
> We use uniform quantization and set the number of quantization levels as per the prior work [1]. However, our ablation study shows that RLEL is not very sensitive to the number of quantization levels if its value is set high. We believe that training of D matrix parameters results in adaptively learning the quantization levels, as RLEL is not very sensitive to the number of quantization levels (Table 1 in Question 1).
>
> There is a potential to adaptively learn the number of quantization levels and non-uniform quantization using the proposed RLEL framework. For example, in Figure 3- step (4), fixed parameters $j$ are used to scale the correlation vector $\hat{C}^j_i$ and find the expected prediction $\hat{y}_i$. These parameters represent quantization levels. One possible approach to learning the quantization levels is to make these parameters trainable. In this case, L1/L2 loss between the expected prediction $\hat{y}_i$ and target labels $y_i$ can be used to train the network.
>
>
> **Changes in the paper:** We have added Sectio A.1.9 to provide the results for different quantization levels and added a discussion on possible future directions for learning quantization levels.

---

> ### Author Response · Authors · 2022-11-19
> **Author response (Part 1)**
>
>
> Thank you for your thoughtful comments and feedback. We have updated the paper and supplemental material to reflect the response below.
>
> **Q1. Deciding the number of quantization levels:**
>
> The number of quantization buckets is treated as a design parameter for binary-encoded labels. Shah et al. [1] showed that the error changes with the number of quantization levels. Fewer levels introduce quantization error, and more levels increase the number of bits in the encoding. They showed a trade-off between these two factors to decide the number of quantization levels.
>
>  Our work focuses on the design space of encoding and decoding functions. Hence we use the same values for the quantization levels ($N$) as BEL [1]. Parameter $N$ tuning can be integrated into hyperparameter tuning or included in the optimization process.
>
> We further analyze the effect of the number of quantization levels for RLEL. The following table shows the NME (Normalized Mean Error) for different values of $N$ for FLD1 benchmark.
> | Quantization levels (N) | NME  |
> |-------------------------|------|
> |                      32 | 3.49 |
> |                      64 | 3.36 |
> |                     128 | 3.36 |
> |                     256 | 3.36 |
> |                     384 | 3.37 |
> |                     512 | 3.37 |
>
> This suggests that the proposed method RLEL is less sensitive to the number of quantization levels for higher values. For RLEL, the decoding matrix that converts the label encodings to the predicted label is also learned during the training (Figure 3). This matrix is of size $M\times N$, where each column represents the weight parameters for one quantization level. One possible reason for the above results is that matrix $D$ adaptively learns the number of quantization levels suitable for this problem.
>
>
>
> **Changes in the paper:** Added results for different quantization levels in Section A.1.8. We have also clarified how we select the value of $N$ in Section 4.1.
>
> <hr>
>
> **Q2. Computational difference between RLEL and BEL**
>
> There is no computational difference between RLEL and BEL for inference.
>
> RLEL and BEL both use the same number of epochs for training. RLEL adds two regularization functions to the loss calculation, which results in more computation (Eq. (3) and Eq. (4)). However, for a batch size of $K$ and label encoding size of $M$, this translates to $K\times K \times M$ additions. For a problem with a batch size of $32$ and label encoding size of $100$, this translates to $\sim10,000$ "Add" operations per iteration, which is significantly less than the number of operations for one training iteration of the ResNet-50 feature extractor [2]. We train BEL and RLEL both on Nvidia RTX 2080 Ti GPU, and we do not observe any notable difference between the training time of both models.
>
> **Changes in the paper**: We have clarified in Section 4.1 that computationally there is no significant difference between RLEL and BEL.
>
> <hr>
>
> **Q3. Explanation of using "2" in Eq. (3)**
>
> Eq. (3) uses max$(0,2 |y_i - y_j | - ||\hat{Z}_i - \hat{Z}_j ||_1)$ as the loss function to encourage $||\hat{Z}_i - \hat{Z}_j ||_1$ value to be greater than $2 |y_i - y_j |$.
>
> Our intuition behind using the scaling parameter $2$ is based on binary-encoded labels. For two adjacent labels (i.e., $|y_i-y_j|=1$), the loss function encourages $||\hat{Z}\_i-\hat{Z}\_{j}||_1$ to be greater than $2$. Here, $\hat{Z}$ is the output label encodings. In the case of binarized label encodings ($-1$ if $Z<0$ and $+1$ if $Z>0$), $||Z_i-Z_j||_1=2$ signifies that two label encodings differ in at least one bit.
>
> We also analyzed the effect of changing this parameter for two benchmarks. The table below shows the impact of changing this scaling parameter for two benchmarks.
>
> | Value of the scaling parameter | NME (FLD1$_s$) | NME ( FLD2$_s$) |
> |--------------------------------|--------------|--------------|
> |                              1 |         4.89 |         4.15 |
> |                              2 |         4.71 |         4.15 |
> |                              3 |         4.83 |         4.20 |
> |                              4 |         4.97 |         4.27 |
> |                              5 |         4.95 |         4.28 |
> |                              6 |         5.06 |         4.41 |
> We observe that the error is higher if the scaling parameter is too low, as label encodings for two adjacent labels can not be discriminated against. If this parameter is set too high, the encoding space is more constrained, and consequently, the performance is degraded.
>
> Based on this intuition and empirical verification for two benchmarks, we use the value $2$ for all benchmarks.
>
> **Changes in the paper:** We have added an ablation study of the effect of the scaling parameter in Section A.1.1. We have clarified the choice of using $2$ as the scaling parameter in Section 3.3.

---

> > ### Comment · Reviewer_ZxJ5 · 2022-11-22
> > **Thanks for the response. I have raised my score.**
> >
> > Thanks for clarifying my questions.

---

### Official Review · Reviewer_vuKX · 2022-10-25

**Confidence:** 4
**Correctness:** 3
**Technical Novelty And Significance:** 4
**Empirical Novelty And Significance:** 4
**Recommendation:** 8

**Clarity, Quality, Novelty And Reproducibility:**

This paper is overall well-written and the proposed method is original. But some notations and analyses are confusing. Source code submitted.

**Strength And Weaknesses:**

Pros:

1. The handcrafted label encoding, though effective, increases the difficulty of the practical deployment of deep regression models. This work proposes an automated label encoding learning method that is much more practical.

2. The experimental results are promising as the proposed method achieves competitive or better performance than manually designed label encodings.

3. The paper is well-organized and mostly written clearly.

Cons:

1. While the presentation is generally good, the methodology is hard to follow. Some notations are not clearly defined. For example, the authors used B, Z, E to denote the label encodings. What are their differences? Why E is the average of E? What is \mathcal{E}(Q_i) on page 5?

2. The 2.1 section (and Figure 2), while important and informative, isn't really clear to me. Why is the bit transition from 1->0 and 0->1 important for label encoding? Also, how does the bit position measures the binary classifier's decision boundary? Personally, I recommend the authors polish this section for a clear explanation of the important properties of label encodings.

3. I noticed that the softmax operator and the cross-entropy loss are used. It suggests the regression network learns a multi-class classifier. While this paper frequently mentioned the binary classifier, I would like to see the difference between these two designs. In particular, it seems a multi-class classifier may reduce the complexity of the encoding space.

4. The authors propose to regularize the matrix D instead of the original label encodings in Eq. (4). While empirically effective, there are no theoretical insights on why it works.


**Summary Of The Paper:**

This paper studies the deep regression problem and proposes a new automated label encoding learning framework along with two regularizers. In specific, the authors relax the assumption of binarized label encodings of the BEL method and propose to search for continuous label embeddings. Moreover, two regularizers are introduced. The first restricts the learned label encoding to preserve the distance of the bits. The second is to enrich the bit transitions. Experiments show the proposed method obtains competitive results to manually designed label encodings.

**Summary Of The Review:**

See above.

---

> ### Author Response · Authors · 2022-11-19
> **Author response ( Part 3 )**
>
> **Q4. Use of matrix D instead of label encodings in Eq. (4)**
>
> We insight into this decision as follows. First, note that the output label encodings are multiplied with $D$ to generate the correlation vector $\hat{C}_i$ (Figure 3). We use the multiclass classification loss between $\hat{C}_i$ and the target labels for training. Due to this, label encodings $E$ and decoding matrix $D$ are related, and use of matrix $D$ proves to be effective for regularizer R2. We further explain this in detail below.
>
> Let $E$ represent an encoding matrix of size $N\times M$. Each row $E_{k:}$ represents the label encoding output when the label is $k$. $D$ is the decoding matrix of size $M\times N$.
>
> Let $\hat{C}\_k$ represent the output correlation row vector of size $1\times N$ when the target label is $k$. Here, $\hat{C}\_{k}$ is obtained by multiplying $E_{k,:}$ with $D$ (Figure 3).
>
> $\hat{C}\_{k} = E_{k,:} D$    (6)
>
> Since we apply softmax on the output vector to find the predicted label (Figure 3), ideally, $\hat{C}^k_{k}$ should have the highest value as the target label value is $k$.
>
> $\therefore \hat{C}^k_{k} > \hat{C}^x_{k} ,\text{ where, } x \neq k, x\in  \{1,2,...,N\}$.
>
> $\therefore E_{k,:} . D_{:,k} > E_{k,:} . D_{:,x},\text{ where, } x \neq k, x\in  \{1,2,...,N\}$ (Using Eq. (6))
>
> Let $\theta_{k,x}$ represents the angle between row vector $E_{k,:}$ and column vector $D_{: ,x}$. This leads to the below equation:
>
> $||E_{k,:}|| ||D_{:,k}|| \text{cos}(\theta_{k,k}) > ||E_{k,:}|| ||D_{:,x}|| \text{cos}(\theta_{k,x}),\text{ where, } x \neq k, x\in  \{1,2,...,N\}$ (7)
>
> Shah et al. [1] used a hand-crafted decoding matrix $D$ with an equal number of $1$s and $0$s in each column for binary-encoded labels. Hence the L2 norm of each column is the same. In label encoding learning, parameters of matrix $D$ are learned during training and are not constrained to have the same L2 norm for each column. However, we observe a similar trend empirically. Figure 9 (in the paper) plots the distribution of $||D_{:,x}||$ for different benchmarks. As shown in the figure, for most benchmarks, we observe a small variance in the distribution of $||D_{:,x}||$. For example, for LFH2 benchmark, the mean and variance of  $||D_{:,x}||$ are $8.18$ and $0.061$, respectively .Based on this intuition and empirical validation, we assume that $||D_{:,x}|| \approx ||D_{:,y}||$ for $x\in[1, N]$ and $y\in[1, N]$ to simplify the analysis.
>
> Using this assumption in Eq. (7) leads to the following inequality:
>
> $\text{cos}(\theta_{k,k}) > \text{cos}(\theta_{k,x}),\text{ where, } x \neq k, x\in  \{1,2,...,N\}$
>
> Thus the cosine similarity between $E_{k,:}$ and $D_{:,k}$ should be the highest to predict label $k$. The optimization process to reduce the loss between the target and prediction will try to maximize this cosine similarity. In the best case, the angle between $E_{k,:}$ and $D_{:,k}$ will be zero, and both vectors are parallel.
>
> This simplification leads to the following relation between $E$ and $D$.
>
> $E_{k,:} = t D_{:,k}$ ,  where $t > 0$
>
> Similarly, $E_{k+1,:} = t' D_{:,k+1}$ , where $t' > 0$
>
> Since $t$ and $t'$ both are positive values, reducing $D_{i,k}-D_{i,k+1}$ also reduces $E_{k,i}-E_{k+1,i}$.
>
> Regularizer rule R2 proposes to regularize the number of decision boundaries by regularizing $\sum_{i=1}^{M} \sum_{j=1}^{N-1} |{E}_{j,i} - E_{j+1,i}|$ as per Eq. (2).
>
> Based on the analysis above, regularizig $\sum_{i=1}^{M} \sum_{j=1}^{N-1} |D_{i,j}-D_{i,j+1} |$ helps with the above goal as $E_{j,i}-E_{j+1,i}$ reduces with $D_{i,j}-D_{i,j+1}$.
>
> **Changes in the paper:** We have added Section A.1.7 to add this theoretical analysis. We have also added a reference to this analysis in the main paper in Section 3.3 (Paragraph on R2).
>
> <hr>
>
> **References**
>
> [1] Deval Shah, Zi Yu Xue, and Tor M. Aamodt. Label encoding for regression networks. In International Conference on Learning Representations, April 2022. URL https://openreview.net/pdf?id=8WawVDdKqlL.

---

> ### Author Response · Authors · 2022-11-19
> **Author response (Part 2)**
>
> **Q3. Clarification on the use of multiclass classifier**
>
> We use binary classifiers in the context of binary-encoded labels. Each bit of the label encoding can be learned using a binary classifier. For example, in Figure 3, binary cross entropy loss between the output code $\hat{Z}_i$ and target binary-encoded labels $B_i$ can be used for training in the case of hand-crafted labels. Similarly, the predicted code can be decoded to a correlation vector ($\hat{C}_i$ in Figure 3), where each value $\hat{C}^j_i$ gives a measure of the probability that the quantized label is equal to $j$. In this case, multiclass classification loss is used between this correlation vector and quantized labels, where the quantized target labels are treated as different classes. Thus, each bit of the label encoding is a binary classifier. However, identifying the label for a predicted multi-bit valued label vector can be treated as a multiclass classification problem.
>
> For hand-crafted labels, label encodings with more than two values (e.g., ternary: $-1$/$0$/$1$) can be used with a multiclass classifier for each bit. However, we explore learning of real-valued label encodings where the properties of real-valued label encodings are based on the properties previously identified to be suitable for binary label encodings. We use multiclass classification loss between the quantized target and the correlation vector $\hat{C}_i$ decoded from label encoding.
>
> **Changes in the paper:** Added clarification in Section 3.3 (Paragraph on “Loss function”)

---

> ### Author Response · Authors · 2022-11-19
> **Author Response (Part 1)**
>
> Thank you for your thoughtful comments and feedback. We have updated the paper and supplemental material to reflect the response below.
>
> **Q1.  Clarification on some notations**
>
> $B$ and $\hat{B}$ represent the target and predicted binary-encoded labels. $B$ is used in the context of using hand-crafted label encodings where it is used as target binary labels for binary classification loss computation. $\mathcal{E}$ is the encoding function. For a training sample with quantized label = $Q_i$, the target binary-encoded label is $B_i=\mathcal{E}(Q_i)$.
>
>
> $\hat{Z}_i$ is the predicted real-valued label encoding for a training example $i$ (Output code in Figure 3). This real-valued label encoding can be further binarized to find the predicted binary code $\hat{B}_i$.
>
>
>
> For regularized label encoding learning (RLEL), $E$ is the learned label encoding. The network learns label encodings throughout the training. Here the *activation values* $\hat{Z}_i$ for a training example $i$ with target label $Q_i=n$ is considered to be learned label encoding for label $n$. However, there can be multiple training examples with target label value equal to $n$, and the output activation values $\hat{Z}_i$ will differ for each of these.
>
> The output activations $\hat{Z}\_i$ for all training examples with target label $Q_i=n$ are averaged to find the learned label encodings $E_{n:}$ for quantized label $n$ in Eq. (1).
>
> We have added a table to summarize all notations used in this work.
> |Notation| Description |
> |-------------------|----------------------------------------------------------------------------------------------------------------------------|
> | $x_i$, $y_i$, $Q_i$     | Input, real-valued target label, and quantized target label  for training example $i$. $y_i \in [1,N]$ and $Q_i \in \{1,2,...,N\}$ |
> | $N$                 | The range of target labels $y_i$; Number of quantization levels for $Q_i$                                                      |
> | $M$                 | Number of bits/values for label encodings                                                                                  |
> | $B_i$, $\hat{B}_i$ | Target and predicted binary label encodings (used for hand-crafted label encodings)                                         |
> | $\mathcal{E}$       | Encoding function (used for hand-crafted label encodings)                                                                   |
> | $\hat{Z}_i$         | Predicted real-valued label encodings; activation values of the output code layer in Figure 3                              |
> | $E$                 | Learned label encodings through RLEL; calculated from $\hat{Z}_i$ for all training examples using Eq. (1)                    |
> | $\hat{C}_i$         | Output correlation vector of length $N$. Here $\hat{C}^j_i$ gives a measure of the probability that the predicted label value is equal to $j$   |
> | $D$                 | Decoding matrix that converts the predicted label encodings to a correlation vector $\hat{C}_i$|
>
> **Changes in the paper:** We have added Table 1 to summarize all the notations (Table 1 in Section 3.1 on Page 5).
>
> <hr>
>
> **Q2. Importance of bit-transitions for label encoding**
>
> Each classifier provides a single bit ($0$/$1$) in the multi-bit binary-encoded label and so must learn the value of this bit for different training examples. The target bit value ($0$/$1$) of the classifier depends upon the target label value. For example, in the Unary encoding (Figure 2a), the target bit for classifier $B^1$ is $1$ when the (real-valued) target label is greater than $1$, else $0$. So this classifier has to learn a decision boundary between $(1,2)$.
>
> Similarly, for the Johnson encoding (Figure 2c), the target bit for classifier $B^3$ is $1$ when the target label is greater than $2$ or less than $7$. In this case, there are two bit-transitions,  $0\rightarrow1$ (between $2$ and $3$) and $1\rightarrow0$ (between $6$ and $7$). This classifier has to learn two decision boundaries in $(2,3)$ and $(6,7)$. As the number of bit-transitions increases, the number of intervals for which the classifier has to learn a separate decision boundary increases, and thus its complexity also increases.
>
> **Changes in the paper:** We have added clarification for this and modified Section 2.1 as per the suggestion (Paragraphs 3 and 4 in Section 2.1).

---

### Decision · Program_Chairs · 2023-01-20

**Decision:**

Accept: notable-top-25%

**Justification For Why Not Higher Score:**

I am not sure how many oral presentations we will have, but perhaps this submission is not that competitive to make an oral presentation.

**Justification For Why Not Lower Score:**

The paper is quite good and studied a very fundamental problem (i.e., regression), so I think many researchers/practitioners may be interested in it.

**Metareview: Summary, Strengths And Weaknesses:**

The paper studied indirect regression based on deep learning that is to perform regression tasks by an encoding-classification-decoding pipeline. This indirect approach is recently shown to be better than direct regression but its label encoding needs to be manually crafted. This paper made it end-to-end with two intuitive and effective regularizations and empirically demonstrated that the learned label encoding can be superior to the crafted one. All reviewers agreed that the novelty and significance are above the bar and thus we should accept it for publication.

**Note From Pc:**

if the above contains the word "oral" or "spotlight" please see: "oral" presentation means -> notable-top-5% and "spotlight" means -> notable-top-25%. As stated in our emails, we are disassociating presentation type from AC recommendations